# Ties Matter: Meta-Evaluating Modern Metrics with Pairwise Accuracy and Tie Calibration

**Daniel Deutsch, George Foster, and Markus Freitag**
Google
{dandeutsch,fosterg,freitag}@google.com

## Abstract

Kendall's $\tau$ is frequently used to meta-evaluate how well machine translation (MT) evaluation metrics score individual translations. Its focus on pairwise score comparisons is intuitive but raises the question of how ties should be handled, a gray area that has motivated different variants in the literature. We demonstrate that, in settings like modern MT meta-evaluation, existing variants have weaknesses arising from their handling of ties, and in some situations can even be gamed. We propose instead to meta-evaluate metrics with a version of pairwise accuracy that gives metrics credit for correctly predicting ties, in combination with a tie calibration procedure that automatically introduces ties into metric scores, enabling fair comparison between metrics that do and do not predict ties. We argue and provide experimental evidence that these modifications lead to fairer ranking-based assessments of metric performance.[1]

## 1 Introduction

Kendall's $\tau$ is a widely used correlation statistic (Kendall, 1938). It is easy to grasp intuitively, being based on pairwise rank ordering. This makes it complementary to other well-known statistics such as Pearson or Spearman.

In the context of machine translation (MT), Kendall plays a key role assessing the performance of evaluation metrics, a process known as meta-evaluation: it has been the main statistic for measuring a metric's ability to score segment-level translations in the Workshop on Machine Translation (WMT) metrics shared tasks over the years (Freitag et al., 2022b, *inter alia*).

Several recent developments in MT—common to other areas of generative AI—have highlighted an important weakness in Kendall's $\tau$, namely how

it deals with ties (§4). First, as MT systems get better, they (1) produce more "perfect" outputs, which get assigned the same score by human raters; and (2) necessitate error-based analyses such as MQM (Lommel et al., 2014; Freitag et al., 2021a), which often produce tied scores due to integer error counts. Second, on the metric side, the use of recently-proposed LLM-based metrics (Kocmi and Federmann, 2023) and metrics that model MQM annotations (Perrella et al., 2022) can also lead to small and discrete score ranges that assign many ties.

In this paper, we examine the problems caused by ties in Kendall's $\tau$, using data from the WMT metrics tasks. We first show that there are simple phenomena that are not handled properly by any of the existing Kendall variants, which mostly differ in how they treat ties (§5.1). We also demonstrate the possibility of gaming the meta-evaluation by exploiting how ties are handled by existing $\tau$'s, resulting in large improvements in certain evaluation settings (§5.2).

We propose instead to meta-evaluate metrics with a version of pairwise accuracy that is robust to these problems, assigning proper credit for correctly predicting ties (§6). Although there is a modification to $\tau$ that is closely related to pairwise accuracy, we argue that the accuracy formulation is easier to interpret, being just the proportion of correctly ranked pairs (including tied pairs).

However, pairwise accuracy comes with its own problem, namely that it can discriminate against metrics that rarely assign ties. To counter this, we also propose an algorithm called *tie calibration* that automatically introduces ties into metric scores in order to optimize its correlation (§7). We argue, and show empirically, that these two modifications result in a fairer assessment of MT metric performance (§8.1).

Finally, we analyze different aspects of pairwise accuracy and tie calibration, including assessing

---

[1]The code to run our proposed methods and reproduce our results can be found at https://github.com/google-research/mt-metrics-eval.

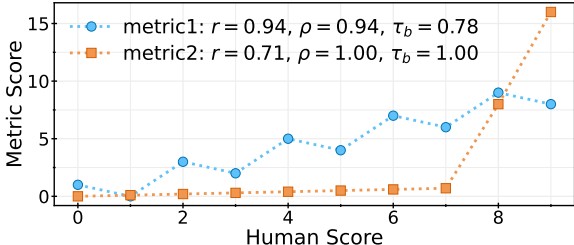

Figure 1: Pearson's $r$, Spearman's $\rho$, and Kendall's $\tau_b$ calculated between hypothetical human scores and metric scores. Lines between data points are shown for visualization purposes.

the generalization of tie calibration across datasets (§8.2), the score ranges where ties are introduced (§8.3), and how more fine-grained statistics can be used to better understand metric behavior (§8.4).

While our experimental setting is limited to MT metrics, our work should be applicable to meta-evaluation for other generative AI metrics with similar characteristics.

## 2 Background & Related Work

We begin by justifying our exclusive focus on ranking-based statistics, like Kendall's $\tau$, then provide some background on MT metric meta-evaluation, and finally contextualize our work by discussing Kendall variants.

### 2.1 Why not Pearson or Spearman?

Pearson's $r$ and Spearman's $\rho$ are two other widely-used correlation coefficients. The Pearson coefficient captures linear correspondence between two input vectors, defined as their covariance divided by the product of their variances. Spearman is equivalent to Pearson applied to the *ranks* of the inputs. As shown in Figure 1, Pearson is complementary to Kendall; it assigns a much higher score to the noisy but globally linear metric1, but a much lower score to the perfectly-ordered but non-linear metric2. Spearman is a compromise, siding with Pearson for metric1 and for Kendall for metric2.

For applications where linear correspondence with a gold standard *and* correct ranking decisions are both important, it is advisable to measure both Pearson and Kendall, as is typically done in the MT evaluations described below.[2]

---

[2]Although we focus on problems with Kendall here, it is worth noting that Pearson has problems of its own, notably sensitivity to outliers (Mathur et al., 2020). For instance, adding the point $(100, 100)$ to metric1 produces an almost perfect correlation of 0.99 compared to 0.82 for Kendall.

## 2.2 Metric Meta-Evaluation

For over 10 years, the Workshop on Machine Translation (WMT) has run a metrics shared task that meta-evaluates automatic metrics. Meta-evaluation quantifies a metric's performance by calculating the agreement or correlation between the metric's scores and human-annotated scores on a large number of translations. In WMT, metrics are meta-evaluated at either the system- or segment-level, as follows.

First, metric and human scores are collected for translations produced by $N$ systems for $M$ source segments. System-level correlations are calculated between the $N$ metric and human scores per system, typically calculated by averaging over the $M$ segment scores. In WMT, the system-level correlation is often Pearson, or more recently, a ranking-based pairwise agreement that is similar to our proposed statistic (§6), except that it does not need to account for ties since ties are very rare at the system-level (Kocmi et al., 2021).

Segment-level correlations evaluate metric scores on individual translations rather than aggregated system scores. They can be calculated in several different ways (see Appendix A for equation definitions):

- **No-Grouping**: Calculate the correlation between the $N \times M$ translation scores

- **Group-by-Item**: Calculate the average correlation between the $N$ translation scores grouped by source segment[3]

- **Group-by-System**: Calculate the average correlation between the $M$ translation scores grouped by system

Segment-level correlations are better than system-level correlations at discriminating between metrics (Freitag et al., 2022b), and they are more closely related to applications where metrics can be used to improve generation, such as Minimum Bayes Risk decoding (Freitag et al., 2022a; Fernandes et al., 2022).

Historically, WMT has evaluated metrics at the segment-level using the group-by-item method, however no-grouping was used in WMT'21 and all three were used in WMT'22. The standard correlation function that is used is some variant of Kendall's $\tau$, described next.

---

[3]"Item" is used to keep the terminology generic so it can be applied to other generation tasks. Here, "item" refers to the source segment.

| Definition | Proposed By | WMT Shared Task Years |
|---|---|---|
| $\tau_a = (C - D)/(C + D + T_h + T_m + T_{hm})$ | Kendall (1938) | - |
| $\tau_b = (C - D)/\sqrt{(C + D + T_h)(C + D + T_m)}$ | Kendall (1945) | 2021–2022 |
| $\tau_c = (C - D)/(n^2(\frac{k-1}{k}))$ | Stuart (1953) | - |
| $\tau_{10} = (C - D - T_m)/(C + D + T_m)$ | Callison-Burch et al. (2010) | 2010–2012, 2017–2020[4] |
| $\tau_{13} = (C - D)/(C + D)$ | Macháček and Bojar (2013) | 2013 |
| $\tau_{14} = (C - D)/(C + D + T_m)$ | Macháček and Bojar (2014) | 2014–2016 |
| $\tau_{eq} = (C + T_{hm} - D - T_h - T_m)/(C + D + T_h + T_m + T_{hm})$ | This work (§6) | - |
| $acc_{eq} = (C + T_{hm})/(C + D + T_h + T_m + T_{hm})$ | This work (§6) | - |

Table 1: Each variant of $\tau$ handles ties differently, and the WMT metrics shared task has not consistently used the same $\tau$ over the years. The $acc_{eq}$ and $\tau_{eq}$ statistics are proposed in this work (§6). See Table 2 for the notation definition for this table.

| Symbol | Description |
|---|---|
| $n$ | The number of inputs |
| $h$ | The vector of human scores |
| $m$ | The vector of metric scores |
| $C$ | The number of concordant pairs |
| $D$ | The number of discordant pairs |
| $T_h$ | The number of pairs tied only in $h$ |
| $T_m$ | The number of pairs tied only in $m$ |
| $T_{hm}$ | The number of pairs tied in both $h$ and $m$ |
| $k$ | The minimum of the number of unique values in $h$ or $m$ |

Table 2: The notation used for defining different $\tau$'s.

## 2.3 The Landscape of Kendall's $\tau$

Kendall's $\tau$ is a ranking-based correlation coefficient. Although there are many different variants of $\tau$, intuitively, it counts how frequently the metric and human scores agree (concordant) or disagree (discordant) on the ranking of all possible pairs of translations. Importantly, there cannot be a tie in either the metric or human score for a pair to be considered concordant or discordant. Each $\tau$ ranges from -1 to 1, with the extremes resulting from the metric and human scores being perfectly discordant/concordant and 0 meaning random chance.

Some variants of $\tau$ are generic and included in libraries like SciPy, whereas others were proposed by WMT metrics shared task organizers and tailored to the application of MT metric meta-evaluation. Table 1 shows the definitions of the different variants of $\tau$ using the notation in Table 2.

The main differences between the variants are how they handle ties. The standard variants, $\tau_b$ and $\tau_c$, are modifications of $\tau_a$ designed to ensure the values can reach -1 and 1 in the presence of ties. In contrast to our proposal, the versions proposed by WMT do not include ties in the human scores, and penalize ties in the metric scores. This is due to the

fact that the metrics shared task organizers either did not want to penalize small differences in metric scores when the human score is tied (Callison-Burch et al., 2010) or only evaluated on pairs that had a large difference in DA score in order to ensure the pair's ranking was reliable (Bojar et al., 2017).

Overall, none of the $\tau$'s directly rewards the metric for correctly predicting ties in the human score. We view our work as a next step in updating the meta-evaluation to account for properties of today's metrics and human scores.

## 3 Analysis Setup

**Datasets** Our analysis is performed on the Multidimensional Quality Metrics (MQM; Lommel et al., 2014; Freitag et al., 2021a) ratings collected by the WMT'22 metrics shared task (Freitag et al., 2022b) for three language pairs: en→de, zh→en, and en→ru. We use the MQM scores as the ground-truth human scores that the automatic metrics' scores are evaluated against. The language pairs have 13-15 systems and around 1300-1900 segments per system with MQM ratings.

**Automatic Metrics** We explore how the choice of meta-evaluation statistic affects the rankings of the primary metric submissions to the WMT'22 shared task, in addition to the recently proposed GEMBA metrics (Kocmi and Federmann, 2023). We also discuss and examine various different metrics in more detail, including the top 2 performing metrics in the WMT'22 shared task, Metric-X and COMET-22 (Rei et al., 2022), in addition to BLEURT-20 (Sellam et al., 2020), MaTESe (Perrella et al., 2022), and GEMBA. The former three metrics are regression-based metrics that predict floating point translation quality scores. MaTESe predicts span-level errors that are combined into an

---

[4] See note about the error in the WMT'17 report in the WMT'18 report (Ma et al., 2018).

| LP | #Segments | Group-by-Item | | |
|---|---|---|---|---|
| | | #Pairs | #Ties | #Zero-Ties |
| en-de | 18k | 120k | 64k (53%) | 48k (40%) |
| zh-en | 28k | 197k | 82k (42%) | 63k (32%) |
| en-ru | 20k | 138k | 61k (44%) | 40k (29%) |

Table 3: The number of segments, pairs, tied pairs, and pairs tied at MQM=0 (error free) across the different WMT'22 language pairs for group-by-item correlations. The statistics for other segment-level correlations can be found in Appendix C.

| Metric | en-de | zh-en | en-ru |
|---|---|---|---|
| Metric-X | 0.7% | 0.2% | 0.5% |
| COMET-22 | 1.3% | 0.1% | 0.5% |
| MaTESe | 71.9% | 39.6% | 80.8% |
| GEMBA-GPT-3.5 | 60.3% | 56.6% | 50.9% |
| GEMBA-GPT-4 | 69.6% | 46.9% | 60.1% |

Table 4: The percent of pairs that are tied when grouping by source segment is drastically different for regression metrics (Metric-X and COMET) versus metrics that effectively act as multi-class classifiers (MaTESe and GEMBA).

$$h = [0, 0, 0, 0, 1, 2]$$
$$m_1 = [0, 0, 0, 0, 2, 1]$$
$$m_2 = [0, 1, 2, 3, 4, 5]$$

| Metric | $\tau_a$ | $\tau_b$ | $\tau_c$ | $\tau_{10}$ | $\tau_{13}$ | $\tau_{14}$ | $\tau_{eq}$ | $acc_{eq}$ |
|---|---|---|---|---|---|---|---|---|
| $m_1$ | .47 | .78 | .29 | .78 | .78 | .78 | .87 | .93 |
| $m_2$ | .60 | .77 | .38 | 1.0 | 1.0 | 1.0 | .20 | .60 |

Figure 2: When considering ties, $m_1$ only incorrectly ranks 1 out of the $\binom{6}{2}$ pairs, whereas $m_2$ incorrectly ranks 6. However, due to how each $\tau$ handles ties, only $acc_{eq}$ and $\tau_{eq}$ strongly prefer $m_1$ over $m_2$. Notably, $\tau_{10}$, $\tau_{13}$, and $\tau_{14}$ are unable to distinguish a perfect metric ($m = h$) from $m_2$. The $acc_{eq}$ and $\tau_{eq}$ statistics are proposed in this work (§6).

overall score based on an error severity weighting. The GEMBA metrics predict quality scores using 0-shot prompting with GPT-3.5 and GPT-4 (Brown et al., 2020). Importantly, the predicted scores from MaTESe and GEMBA tend to come from a small set of values rather than a large range of possible floating point scores, which has significant implications for the number of ties they predict (see §4) and how they are treated by different variants of $\tau$.

# 4 Why Ties are Important

There are several motivations for incorporating ties into a ranking-based meta-evaluation statistic like Kendall's $\tau$.

First, ties in human scores from recent WMT shared tasks are much more trustworthy than they were previously. Since WMT'20, the human scores are MQM scores instead of direct assessment (DA) scores. The MQM scores come from expert translators and are more reliable than the crowdsourced DA scores. As such, ties (or minor differences) between scores are more likely representative of actual ties (or minor differences) in translation quality.

Second, ties in MQM scores are very common. For instance, up to 53% of possible pairs in en-de have tied MQM scores (see Table 3), the majority of which have MQM scores of 0, meaning there are no errors in the translations. As the quality of MT systems improves, the number of tied translations is likely to increase since there will be fewer differences between systems. If ties in the MQM scores are removed from the meta-evaluation (as is done by some Kendall variants), we throw away a valuable metric quality signal and lose the ability to discriminate between metrics that reliably detect ties and those that do not (see next section).

Finally, recently proposed metrics, such as MaTESe or those based on large language mod-

els (GEMBA) predict a large number of ties (see Table 4). These metrics should be directly rewarded for correctly predicting ties in the human scores, which is not the case with existing Kendall variants.

# 5 Shortcomings of Kendall's Variants

The way ties are handled by existing variants of Kendall's $\tau$ introduces blind spots in the meta-evaluation and opens the door for metrics to exploit $\tau$-specific properties to improve correlations. We demonstrate the shortcomings of existing $\tau$'s through a motivational example and experimental analysis.

## 5.1 A Motivating Example

Due to how existing $\tau$'s handle ties, they are unable to discriminate between metrics that accurately predict ties and those that do not. Figure 2 contains an example of such an instance.

When considering ties, metric $m_1$ only incorrectly ranks 1 out of the 15 possible pairs, whereas $m_2$ incorrectly ranks 6 pairs. However, because existing $\tau$'s do not give credit to metrics for correctly predicting ties, the correlation coefficients

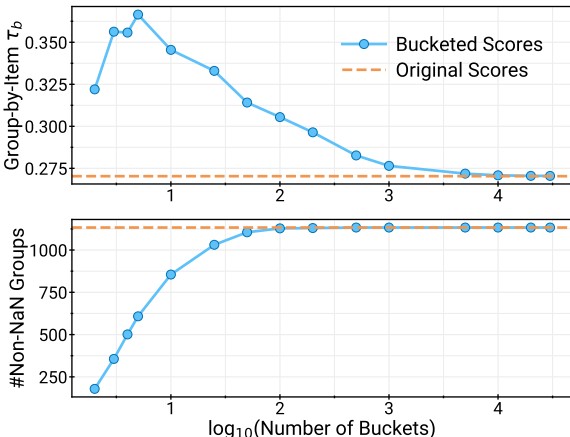

Figure 3: Dividing the Metric-X scores into equal width buckets can increase the group-by-item correlation by a large margin. However, at the same time, the number of groups used in the correlation (with non-NaN scores) decreases, meaning the corresponding correlations are not fairly comparable since they are computed on different sets of data.

either consider $m_1$ to be either approximately equal or worse than $m_2$. This blind spot of existing $\tau$'s means they are inadequate for meta-evaluating metrics in the presence of ties.

## 5.2 The NaN Problem

Another consequence of how the $\tau$ correlations handle ties is what we refer to as the "NaN problem." In the event that either the metric or human scores are a constant vector (therefore, all pairs are tied), many of the $\tau$ values are not defined, or NaN. When the segment-level correlation is calculated by grouping by either item or system and one of the groups' correlations is NaN, the correlation is removed from the average in practice. This happens most often when grouping by item because the size of the input vectors is the number of systems, $N$, which is generally rather small ($\approx$15).

A metric could take advantage of this property of the segment-level correlation by introducing ties for difficult-to-score groups, resulting in NaN scores. This has the effect of removing the challenging groups from the meta-evaluation, resulting in higher correlations.[5,6] Indeed, we find that this is possible.

To introduce ties, we mapped Metric-X's scores

---

[5]Another possibility would be to assign a neutral correlation value of 0. However, this has the disadvantage of penalizing metrics that assign ties when all human scores are also tied or are close to being tied.

[6]Note that both Pearson and Spearman are also NaN for constant vectors and therefore are also susceptible to gaming.

to integers by assigning each score to an equal-width bucket. This bucketing results in ties in challenging pairs because similar quality translations likely have close metric scores, so when the scores are converted to integer buckets, their scores become the same value. Figure 3 plots the group-by-item $\tau_b$ (the $\tau$ coefficient used in WMT'22) and the number of non-NaN groups as a function of the number of buckets.

When the number of buckets is small, the number of non-NaN segments is reduced, and the resulting correlations improve over the original values by very large margins. Because the correlations with different numbers of buckets are computed over different non-NaN subsets of the full dataset, their values are not fairly comparable. Indeed, in §8, we demonstrate that WMT'22 metrics submissions were evaluated on different non-NaN groups, and directly comparing their correlations leads to erroneous conclusions.

A metric could have taken advantage of the NaN problem in order to game the WMT'22 metrics shared task since the number of non-NaN segments is not taken into account in the metric meta-evaluation. A method for handling ties that made correlations for constant vectors well defined would close this loophole.

## 6 Evaluating with Pairwise Accuracy

Instead of using Kendall's $\tau$ as the ranking-based meta-evaluation statistic, we propose to use a version of pairwise accuracy that includes ties. We define the pairwise accuracy to be the proportion of all pairs that the metric either ranks correctly or correctly predicts are tied. The equation for our proposal, denoted $acc_{eq}$ ("eq" for equality, as in ties), is included in Table 1. This statistic now directly incorporates ties in the human and metric scores.

Although there is a modification of Kendall's $\tau$ that corresponds to $acc_{eq}$ (denoted $\tau_{eq}$ in Table 1), we advocate for reporting accuracy instead. Accuracy is more intuitive than $\tau$ since its value is between 0 and 1 and it can be read as the proportion of pairs that the metric correctly ranks/predicts as ties. This stands in contrast to $\tau$ that is between -1 and 1, which does not have an easy-to-communicate interpretation. Pairwise accuracy has the additional benefit of aligning how metrics are meta-evaluated at the system- and segment-levels (Kocmi et al., 2021). The results related to metric rankings in this

| Metric | Definition |
|---|---|
| ties$_{\text{precision}}$ | $T_{hm}/(T_{hm} + T_m)$ |
| ties$_{\text{recall}}$ | $T_{hm}/(T_{hm} + T_h)$ |
| correct-rank$_{\text{precision}}$ | $C/(C + D + T_h)$ |
| correct-rank$_{\text{recall}}$ | $C/(C + D + T_m)$ |

Table 5: Definitions of precision and recall on correctly predicting tied pairs or the correct ranking of non-tied pairs. See Table 2 for the notation definition.

work apply equally to acc$_{\text{eq}}$ and $\tau_{\text{eq}}$.

Pairwise accuracy (and $\tau_{\text{eq}}$) does not suffer from the same issues as the $\tau$'s that were presented in §5: acc$_{\text{eq}}$ strongly prefers $m_1$, the metric with fewer incorrectly ranked pairs (Figure 2; §5.1). Because its value is never NaN, it does not suffer from the NaN problem (§5.2); all examples are always used for evaluation.

### 6.1 Evaluating Ties and Non-Ties

Pairwise accuracy effectively evaluates the automatic metrics as 3-way classifiers that decide between predicting a tie or one of the two possible rankings for each pair. This formulation nicely allows for further decomposition into class-specific precision, recall, and F$_1$, which can be used to further understand metric performance. Class-specific evaluations help to address a potential class imbalance problem between tied and non-tied pairs that may be hidden by accuracy.

Table 5 contains the definitions of precision and recall with respect to "ties" and "correct ranking." The "ties" statistics calculate the precision of the metric when predicting a tie and its recall of human ties. The "correct ranking" statistics calculate the proportion of correctly ranked pairs out of all pairs it predicts are not tied and the proportion of all human non-tied pairs correctly ranked by the metric. These additional statistics help provide a more holistic view of metric performance.

## 7 Tie Calibration

Although we argue that acc$_{\text{eq}}$ properly addresses ties in human and metric scores, some metrics do not frequently predict exact ties between translations. Regression metrics, such as BLEURT (Sellam et al., 2020) and COMET (Rei et al., 2020), practically never predict tied scores for two different translations (see Table 4), so they will not able to correctly predict a tie in the human score, putting them at a disadvantage. This is undesirable because

it prevents a fair comparison between metrics that do and do not predict ties.

To address this shortcoming, we propose an algorithm called *tie calibration* for automatically introducing ties into metric scores so that metrics that do and do not predict ties can be fairly compared. The algorithm is based on the intuition that, although regression metrics do not frequently predict ties, the difference between two translations' scores is sometimes small enough to be considered a tie.

Tie calibration searches for an $\epsilon$ value that maximizes a rank-based correlation statistic (e.g., $\tau$ or acc$_{\text{eq}}$) such that any two translations with a difference in score less than $\epsilon$ is considered to be a tie.[7] Our implementation considers all possible differences between the $\binom{n}{2}$ pairs of translations as candidates for $\epsilon$ and selects the one that maximizes the desired ranking-based statistic. The algorithm runs in $\mathcal{O}(n^2 \log n)$, where $n$ is the number of translations.[8] Detailed psuedocode for tie calibration is included in Appendix D.

Because tie calibration introduces an optimal number of tie predictions, metrics are not penalized for under-predicting ties, and therefore metrics that do and do not predict ties can be fairly compared. An added benefit of tie calibration is that the resulting optimal $\epsilon$ improves the interpretability of metric scores. Its value can be understood as the threshold for which a difference in metric scores should be considered significant (at least with respect to a specific dataset; see §8.2).

Henceforth we use $*$ to denote a statistic that has been calculated with tie calibration (e.g., acc$_{\text{eq}}^*$) and $\epsilon^*$ the optimal tie threshold found by the algorithm.

**Discussion.** In principle, tie calibration can be used to find an optimal value of any correlation statistic in which the presence of ties changes the value, acc$_{\text{eq}}$ being one of them. However, care needs to be taken to ensure that the statistic handles ties in a desirable way. For example, $\tau_{13}$ omits all ties from its formula, so tie calibration could convert a discordant pair into a tie to improve the value of $\tau_{13}$, which, if the human scores are not tied, is undesirable (acc$_{\text{eq}}$ would not reward this change). The combination of tie calibration and a

---

[7]We experimented with relative differences between scores and found little difference compared to absolute differences.

[8]In practice, when $n$ is large, we downsample the number of pairs to consider when searching for $\epsilon$, which significantly improves runtime. Experimentally, this appears to be a rather good approximation (see Appendix E).

| Metric | $\tau_b$ | $\tau_{10}$ | $\text{acc}^*_{\text{eq}}$ | $\epsilon^*$ |
|---|---|---|---|---|
| Metric-X | 0.270 ( 4) | 0.381 ( 1) | 0.605 ( 1) | 0.04 |
| UniTE | 0.278 ( 3) | 0.322 ( 3) | 0.595 ( 2) | 0.14 |
| COMET-22 | 0.258 ( 5) | 0.366 ( 2) | 0.594 ( 3) | 0.11 |
| MaTESe | 0.281 ( 2) | -0.459 (16) | 0.582 ( 4) | 0.00 |
| UniTE-src | 0.205 ( 9) | 0.221 ( 8) | 0.582 ( 5) | 0.12 |
| GEMBA-GPT-4 | 0.322 ( 1) | -0.367 (15) | 0.573 ( 6) | 4.00 |
| MaTESe-QE | 0.234 ( 7) | -0.573 (17) | 0.572 ( 7) | 0.00 |
| COMETKiwi | 0.181 (12) | 0.254 ( 6) | 0.572 ( 8) | 0.16 |
| BLEURT-20 | 0.254 ( 6) | 0.289 ( 4) | 0.568 ( 9) | 0.09 |
| MS-COMET-22 | 0.169 (13) | 0.241 ( 7) | 0.565 (10) | 4.65 |
| COMET-QE | 0.138 (14) | 0.179 (10) | 0.555 (11) | 0.01 |
| SEScore | 0.182 (10) | 0.269 ( 5) | 0.554 (12) | 1.30 |
| MS-COMET-QE-22 | 0.080 (16) | 0.116 (12) | 0.550 (13) | 6.50 |
| HWTSC-Teacher-Sim | 0.106 (15) | 0.123 (11) | 0.545 (14) | 0.34 |
| GEMBA-GPT-3.5 | 0.209 ( 8) | -0.344 (14) | 0.545 (15) | 15.00 |
| MEE4 | 0.182 (11) | 0.201 ( 9) | 0.539 (16) | 0.13 |
| REUSE | -0.074 (18) | -0.134 (13) | 0.534 (17) | 0.47 |
| Constant-Metric | 0.000 (17) | -1.000 (18) | 0.534 (18) | 0.00 |

Table 6: The correlations (and ranks) of the metrics as evaluated by $\tau_b$, $\tau_{10}$, and $\text{acc}_{\text{eq}}$ with tie calibration, denoted $\text{acc}^*_{\text{eq}}$, using the group-by-item segment-level correlation on the WMT'22 en-de dataset. $\epsilon^*$ is the optimal threshold found by tie calibration.

statistic that does not properly handle ties may lead to unexpected results.

# 8 Analysis

In this section, we analyze several different aspects related to our proposal of pairwise accuracy and tie calibration. We address the following questions:

- §8.1: How does the choice of meta-evaluation statistic affect metric ranking?

- §8.2: How does the selected value of $\epsilon$ generalize across datasets?

- §8.3: Does the selected $\epsilon$ value introduce ties uniformly across score values for a metric?

- §8.4: What insights can be drawn from evaluating metrics on predicting tied versus non-tied pairs?

## 8.1 Comparing Metric Rankings

Table 6 shows group-by-item correlations calculated with various $\tau$'s and pairwise accuracy. We also report the performance of a "constant metric" that predicts a tie for every pair as a baseline comparison. From the existing $\tau$'s, we report $\tau_b$ and $\tau_{10}$ since they are the most recent and used most frequently by WMT.[9] Clearly, the choice of meta-evaluation statistic significantly affects the metric rankings, with the largest changes happening to

[9]See Appendix C for the results for each language pair, type of segment-level correlation, and correlation statistic.

MaTESe and GEMBA, the two metrics that output the most ties.

Under $\tau_b$, GEMBA-GPT-4 and MaTESe are the top ranked metrics. However, this result can be partially explained by the NaN problem (§5.2). MaTESe's correlation is calculated on 773 non-NaN segments, compared to 1133 for Metric-X. When both metrics are evaluated on the same 773 segments, Metric-X's correlation is higher (0.296 versus 0.281). This result highlights how correlations calculated on different source segments cannot be fairly compared.

If $\tau_{10}$ is used to rank the metrics, MaTESe and GEMBA fall to the bottom at of the ranking. This result can be explained by the fact that $\tau_{10}$ is systematically biased against metrics that output a large number of ties because ties are penalized as if they are discordant pairs. Predicting a tie can only decrease a $\tau_{10}$ correlation. In fact, the $\tau_{10}$ values of MaTESe and GEMBA can be improved by large margins simply by randomly breaking all ties since around half of the pairs will now become concordant, while the other half remain penalized as they were before. For example, randomly breaking ties improves MaTESe and GEMBA-GPT-3.5's correlations by around 0.5-0.6 points (MaTESe: $-0.459$ to $\approx 0.15$, GEMBA: $-0.344$ to $\approx 0.15$). In contrast, COMET-22's correlation only improves by $\approx 0.005$ due to the fact that it predicts few ties (see Table 4).

In contrast, when the metrics are ranked by $\text{acc}_{\text{eq}}$ with tie calibration, denoted $\text{acc}^*_{\text{eq}}$, MaTESe and the GEMBA metrics are ranked 4th, 6th, and 15th. Because $\tau_{\text{eq}}$ and $\text{acc}_{\text{eq}}$ are never NaN, all values are fairly comparable. Further, there is no systematic bias for or against ties; Randomly breaking or introducing ties runs the risk of changing a correct prediction of a tie or concordant pair into a discordant pair or an incorrect tie prediction. Clearly, the choice of correlation statistic matters, and we argue that $\text{acc}^*_{\text{eq}}$ is the most fair and reliable method compared to the $\tau$ variants.

## 8.2 Generalization of Epsilon

The previous analysis selected $\epsilon^*$ on the same dataset that is used to rank the metrics. Here, we examine what happens if the $\epsilon$ value is selected on a held-out dataset. For this analysis, the MQM ratings from the WMT'21 metrics shared task (Freitag et al., 2021b) are used as a held-out set.

Figure 4 shows the different $\epsilon^*$ and $\text{acc}^*_{\text{eq}}$ values for BLEURT-20 when $\epsilon$ is selected on one dataset

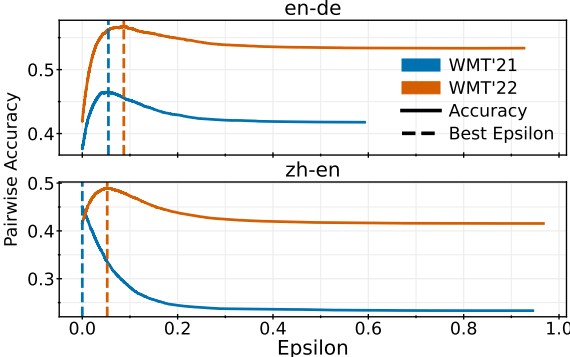

Figure 4: The generalization of the selected $\epsilon^*$ (dashed line) across datasets appears to depend on specific properties of the datasets. We suspect if the number of ties in the datasets is very different (as in zh-en), the $\epsilon$ is less likely to generalize well.

and applied to the other for en-de and zh-en. For en-de, the epsilon value changes by 0.03, and the $acc_{eq}$ calculated on the held-out $\epsilon$ changes by relative 2%, suggesting the results are rather stable.

However, zh-en behaves quite differently. From the plots, it is clear that for WMT'21, there is almost never an incentive to predict a tie, as evidenced by the very low $\epsilon^*$, and the corresponding $\epsilon^*$ does not generalize well to WMT'22 (or vice versa). Our hypothesis is that this result is due to the fact that the WMT'21 zh-en data has far fewer ties than the WMT'22 data (23% versus 41%).

These results indicate that $\epsilon^*$ values are not likely to generalize across dissimilar datasets under current metrics. Such a property would be desirable—and an interesting challenge for metric developers—since it would make score differences more interpretable. However, we argue that treating $\epsilon^*$ as a latent variable calibrated on the current test set allows for fair comparisons of metrics even in the absence of this property. Other evaluation protocols have also involved optimizations on the test set, for example using an oracle sentence segmenter to evaluate MT for speech (Matusov et al., 2005).

### 8.3 Where are Ties Introduced?

Since most of the human score ties are for error free translations (see Table 3), it is worth understanding if the tie threshold introduces ties for high scoring translations to predict error-free translation ties or if the ties are introduced more uniformly across the score distribution.

Figure 5 plots the distribution of the average score per pair where ties are introduced by $\epsilon^*$ for Metric-X on the WMT'22 zh-en dataset. In comparison to the distribution of all pairs' average scores, the tied distribution is skewed toward higher predicted scores. Since Metric-X has a relatively strong correlation to MQM scores, this suggests that the newly introduced ties mostly predict perfect translations, which are assigned high scores according to the metric. An extension of our tie calibration procedure could first identify a threshold to predict a perfect translation, then run tie calibration on the remaining pairs.

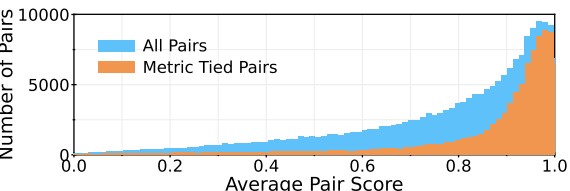

Figure 5: The distribution of average pair scores where ties are introduced for Metric-X on WMT'22 zh-en using $\epsilon^*$ as the tie threshold is skewed right with respect to the distribution of all pairs, suggesting the $\epsilon$ is biased toward introducing ties to predict perfect translations.

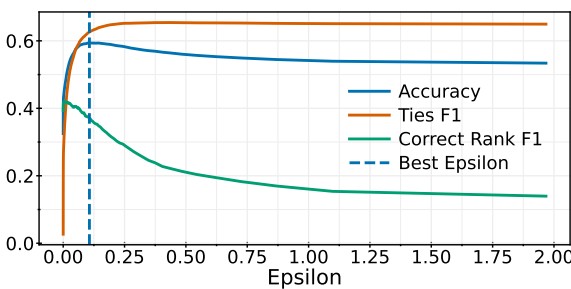

Figure 6: The $F_1$ scores for predicting ties or correct pair rankings for COMET-22 on WMT'22 en-de demonstrate the metric is better at predicting ties than correct pair rankings.

### 8.4 Class-Specific Statistics

Figure 6 plots the ties-$F_1$, correct-rank-$F_1$ (see §6.1), and pairwise accuracy for COMET-22 on en-de. The ties-$F_1$ is much higher than the correct-rank-$F_1$ for almost every $\epsilon$, demonstrating that the metric more reliably predicts tied pairs than the correct rank for non-tied pairs. This is likely due to the fact that the number of perfect translations is large, and the $\epsilon$ values are biased toward introducing ties to predict perfect translations (§8.3).

If a statistic other than pairwise accuracy is better aligned to how a metric is being used in practice, the tie calibration procedure can be used to select an $\epsilon$ that strikes the desired balance of performance with respect to the class-specific statistics.

# 9 Conclusion

In this work, we demonstrated the importance of taking ties into account when calculating rank-based correlation statistics. We argued existing variants of Kendall's $\tau$ are inadequate for the current state of meta-evaluation. We advocated to instead use pairwise accuracy, which rewards metrics for both predicting correct pair rankings and correctly predicting ties, in combination with a tie calibration procedure that allows for comparing metrics that do and do not predict ties. Although our experiments were specific to MT, the methods proposed are generally applicable to any metric meta-evaluation in NLP.

## Limitations

The tie calibration algorithm introduced in §7 makes an assumption that absolute differences in metric scores reflect the same amount of change in quality for any value of the metric. That is, the difference in predicted quality between translations with scores 0.2 and 0.1 is the same as with scores 100.2 and 100.1. An alternative version of the tie calibration algorithm could introduce ties based on relative differences between scores instead of absolute differences. We experimented with relative differences and did not see a significant different in results. However, it may be that a metric that we did not experiment with performs better with relative $\epsilon$ instead of an absolute difference.

Since the tie decision operates at the pair level, the $\epsilon$ value does not induce a global ordering of translations. For example, if there are scores 1, 2, and 3 with $\epsilon = 1$, pairs (1, 2) and (2, 3) are tied but (1, 3) is not. A similar limitation can also be observed in pairwise statistical significance testing.

Finally, although we argue that our meta-evaluation proposal is more fair, we are unaware of any way to prove that this is true. Instead, we rely on experimental results and the fact that our proposals are not susceptible to known issues with existing methodologies.

## Acknowledgments

The authors would like to thank Juraj Juraska, Mara Finkelstein, Ricardo Rei, Chi-kiu Lo, Tom Kocmi, and Alon Lavie for their helpful discussions and feedback related to this work.

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

# A Correlation Definitions

This section more explicitly defines the three different types of segment-level correlations.

Let $h_{ij}$ and $m_{ij}$ denote the human and metric scores for the translation produced by system $i \in 1, \ldots, N$ on source segment $j \in 1, \ldots, M$. Define $\mathrm{Corr}(\cdot)$ to be a correlation coefficient, such as Pearson's $r$, Spearman's $\rho$, Kendall's $\tau$, or any such function that calculates an agreement score over a set of paired observations, like the pairwise accuracy statistic proposed in this work. There are three different segment-level correlations that can be computed.

1. No-Grouping:

$$\mathrm{Corr}\left(\{(h_{ij}, m_{ij})\}_{i=1,j=1}^{N,M}\right) \quad (1)$$

2. Group-by-Item:

$$\frac{1}{M}\sum_{j=1}^{M}\mathrm{Corr}\left(\{(h_{ij}, m_{ij})\}_{i=1}^{N}\right) \quad (2)$$

3. Group-by-System:

$$\frac{1}{N}\sum_{i=1}^{N}\mathrm{Corr}\left(\{(h_{ij}, m_{ij})\}_{j=1}^{M}\right) \quad (3)$$

|  | Metric | | |
|---|---|---|---|
| Notation | < | = | > |
| Human < | $C$ | $T_m$ | $D$ |
| Human = | $T_h$ | $T_{hm}$ | $T_h$ |
| Human > | $D$ | $T_m$ | $C$ |

Table 7: A mapping between the notation in this paper and the tabular notation from WMT'14.

|  | Metric | | |
|---|---|---|---|
| $\tau_{10}$ | < | = | > |
| Human < | 1 | -1 | -1 |
| Human = | X | X | X |
| Human > | -1 | -1 | 1 |

|  | Metric | | |
|---|---|---|---|
| $\tau_{13}$ | < | = | > |
| Human < | 1 | X | -1 |
| Human = | X | X | X |
| Human > | -1 | X | 1 |

|  | Metric | | |
|---|---|---|---|
| $\tau_{14}$ | < | = | > |
| Human < | 1 | 0 | -1 |
| Human = | X | X | X |
| Human > | -1 | 0 | 1 |

Table 8: The tabular versions $\tau_{10}$, $\tau_{13}$ and $\tau_{14}$.

|  | Metric | | |
|---|---|---|---|
| $\tau_{eq}$ | < | = | > |
| Human < | 1 | -1 | -1 |
| Human = | -1 | 1 | -1 |
| Human > | -1 | -1 | 1 |

|  | Metric | | |
|---|---|---|---|
| $\mathrm{acc}_{eq}$ | < | = | > |
| Human < | 1 | 0 | 0 |
| Human = | 0 | 1 | 0 |
| Human > | 0 | 0 | 1 |

Table 9: The tabular versions of $\tau_{eq}$ and $\mathrm{acc}_{eq}$.

## C  Additional Results

Table 10 contains more statistics related to the number of tied pairs in the WMT'22 MQM scores, including the number of pairs that are tied with a score of 0 (i.e., an error free translation).

The full correlation results and metric ranks according to the different $\tau$s across different language pairs and segment-level correlations is included in this section. See Table 11 for the listing of the individual tables.

## D  Tie Calibration Psuedocode

Algorithm 1 contains the pseudocode for the tie calibration procedure (§7) when applied to two vectors of human and metric scores. The algorithm runs in $\mathcal{O}(n^2 \log n)$ where $n$ is the number of scored translations. The bottleneck is sorting all of the $\binom{n}{2}$ possible pairs. When $\binom{n}{2}$ is too large, we approximate the search for the optimal $\epsilon$ by downsampling the number of pairs. See Appendix E for an analysis of how lossy this approximation is.

In practice, the tie calibration is applied to matrices of human and metric scores, where each row corresponds to a group (see §2). The algorithm is very similar to Algorithm 1 except there is additional bookkeeping required to match each $(i, j)$ pair to the group that it came from. The extra bookkeeping only adds an $\mathcal{O}(1)$ overhead.

## E  Epsilon Search Approximation

Finding the exact $\epsilon$ value that maximizes pairwise accuracy requires considering all possible $\binom{n}{2}$ choices of $\epsilon$. For specific segment-level correlations, such as the no-grouping variant, $\binom{n}{2}$ can be prohibitively large, on the order of hundreds of millions of pairs (see Table 3).

## B  WMT Tabular Notation

WMT'14 (Macháček and Bojar, 2014) developed a tabular notation to describe how Kendall's $\tau$ was calculated. For completeness, we include a mapping of the notation from this work in Table 2 to the tabular notation in Table 7. The tabular versions of $\tau_{10}$, $\tau_{13}$, and $\tau_{14}$ are reproduced in Table 8. The tabular versions of $\tau_{eq}$ and $\mathrm{acc}_{eq}$ are included in Table 9.

Re-using the notation from WMT'14, a $\tau$ value can be computed using the tabular notation via the following equation:

$$\tau = \frac{\sum_{\substack{h,m \in \{<,=,>\} \\ C_{h,m} \neq X}} C_{h,m} |S_{h,m}|}{\sum_{\substack{h,m \in \{<,=,>\} \\ C_{h,m} \neq X}} |S_{h,m}|} \quad (4)$$

$C_{h,m}$ is defined as the coefficient in the tabular notation and $S_{h,m}$ is the number of pairs that fall into the corresponding bucket.

| LP | #Translations | No Grouping | | | Group-by-Item | | | Group-by-System | | |
|---|---|---|---|---|---|---|---|---|---|---|
| | | #Pairs | #Ties | #Zero-Ties | #Pairs | #Ties | #Zero-Ties | #Pairs | #Ties | #Zero-Ties |
| en-de | 18k | 169m | 58m (34%) | 49m (29%) | 120k | 64k (53%) | 48k (40%) | 12m | 4m (35%) | 4m (29%) |
| zh-en | 28k | 395m | 103m (26%) | 87m (22%) | 197k | 82k (42%) | 63k (32%) | 26m | 7m (26%) | 6m (22%) |
| en-ru | 20k | 195m | 44m (22%) | 35m (18%) | 138k | 61k (44%) | 40k (29%) | 13m | 3m (23%) | 2m (19%) |

Table 10: The number of translations, pairs, tied pairs, and pairs tied at MQM=0 (perfect translations) across the different WMT'22 language pairs and segment-level correlations.

| LP | Correlation | Table |
|---|---|---|
| | No-Grouping | Table 12 |
| en-de | Group-by-Item | Table 13 |
| | Group-by-System | Table 14 |
| | No-Grouping | Table 15 |
| en-ru | Group-by-Item | Table 16 |
| | Group-by-System | Table 17 |
| | No-Grouping | Table 18 |
| zh-en | Group-by-Item | Table 19 |
| | Group-by-System | Table 20 |

Table 11: Pointers to the full correlation and metric ranking results under different $\tau$s for each language pair and type of segment-level correlation.

When the number of pairs is too large, we instead find an approximate best $\epsilon$ by sampling from all possible pairs. Figure 7 plots the $\epsilon^*$ values calculated on a subset of the data and the value of $\text{acc}_{\text{eq}}$ for those $\epsilon^*$ values. Even with as little as 10% of the possible pairs, the approximations are quite precise. The largest observed differences over 30 iterations for $\epsilon^*$ were 2.5e-3 and for $\text{acc}_{\text{eq}}$ were 4.3e-5. Overall, downsampling appears to be a safe approximation to improve the run time of the tie calibration algorithm.

## F   Unbabel Normalization

The experiments in this paper calculate MQM scores for translations using the normalization technique advocated for by Google: a translation's MQM score is the sum of the weights of each of the errors. The alternative method used by Unbabel normalizes the sum of error weights by the length of the translation. The Unbabel normalization will thus result in fewer human score ties than the Google normalization.

We repeated the analysis from §8.1 using the Unbabel normalization method and calculated the rankings of the different metrics under $\text{acc}_{\text{eq}}$ and $\tau$ variants for the en-ru language pair. The results

---

**Algorithm 1** An $\mathcal{O}(n^2 \log n)$ algorithm that introduces metric ties to select an optimal $\tau$ value.

```
 1: function TIECALIBRATION(h, m, τ)
 2:     C, D, T_h, T_m, T_hm ← SUFFSTATS(h, m, 0)
 3:     P = [(i, j) : i, j = 1, . . . , n; i < j]
 4:     Sort P by |m_i − m_j|
 5:     τ* ← −∞
 6:     τ_curr ← τ(C, D, T_h, T_m, T_hm)
 7:     ε_curr ← 0
 8:     for (i, j) ∈ P do
 9:         if |m_i − m_j| ≠ ε_curr then
10:             τ* ← max(τ*, τ_curr)
11:         Remove (i, j) from C, D, T_h, T_m, T_hm
12:         if h_i = h_j then
13:             T_hm ← T_hm + 1
14:         else
15:             T_m ← T_m + 1
16:         τ_curr ← τ(C, D, T_h, T_m, T_hm)
17:         ε_curr ← |m_i − m_j|
18:     τ* ← max(τ*, τ_curr)
19:     return τ*
20: end function
21: function SUFFSTATS(h, m, ε)
22:     C, D, T_h, T_m, T_hm ← 0, 0, 0, 0, 0
23:     for (i, j) ∈ {(i, j) : i, j = 1, . . . , n; i < j} do
24:         if h_i = h_j and |m_i − m_j| ≤ ε then
25:             T_hm ← T_hm + 1
26:         else if h_i = h_j then
27:             T_h ← T_h + 1
28:         else if |m_i − m_j| ≤ ε then
29:             T_m ← T_m + 1
30:         else if Sign(h_i − h_j) = Sign(m_i − m_j) then
31:             C ← C + 1
32:         else
33:             D ← D + 1
34:     return C, D, T_h, T_m, T_hm
35: end function
```

for the group-by-item segment-level correlation are shown in Table 21.

Overall, the fewer ties did not make a significant impact on whether or not it was possible to demonstrate that the meta-evaluation statistics are biased toward or against ties. For instance, $\tau_b$ favors metrics with ties, such as GEMBA-GPT-4, and $\tau_{10}$ is still biased against metrics that predict ties. We suspect this is due to the fact that the majority of ties occur for perfect translations, which will remain tied in either normalization method. Further, the number of non-perfect ties (MQM score of 0) only decreased by 7% (from 44% to 37%). Therefore,

| Metric | $\tau_a$ | $\tau_b$ | $\tau_c$ | $\tau_{10}$ | $\tau_{13}$ | $\tau_{14}$ | $acc_{eq}$ | $acc^*_{eq}$ | $\epsilon^*$ |
|---|---|---|---|---|---|---|---|---|---|
| Metric-X | 0.289 ( 2) | 0.356 ( 2) | 0.293 ( 2) | 0.440 ( 2) | 0.440 ( 4) | 0.440 ( 2) | 0.473 ( 4) | 0.525 ( 1) | 0.05 |
| UniTE | 0.288 ( 3) | 0.356 ( 3) | 0.292 ( 3) | 0.438 ( 3) | 0.438 ( 5) | 0.438 ( 3) | 0.473 ( 5) | 0.519 ( 2) | 0.11 |
| COMET-22 | 0.292 ( 1) | 0.361 ( 1) | 0.296 ( 1) | 0.445 ( 1) | 0.445 ( 3) | 0.445 ( 1) | 0.475 ( 3) | 0.518 ( 3) | 0.14 |
| MaTESe | 0.170 (14) | 0.323 ( 6) | 0.181 (14) | -0.241 (16) | 0.516 ( 2) | 0.258 (14) | 0.500 ( 1) | 0.494 ( 4) | 0.00 |
| GEMBA-GPT-4 | 0.199 (11) | 0.347 ( 4) | 0.214 (11) | -0.153 (15) | 0.555 ( 1) | 0.302 (11) | 0.481 ( 2) | 0.493 ( 5) | 4.00 |
| BLEURT-20 | 0.274 ( 4) | 0.338 ( 5) | 0.278 ( 4) | 0.417 ( 4) | 0.417 ( 8) | 0.417 ( 4) | 0.466 ( 6) | 0.490 ( 6) | 0.06 |
| MS-COMET-22 | 0.225 ( 7) | 0.277 (10) | 0.228 ( 7) | 0.342 ( 8) | 0.342 (11) | 0.342 ( 7) | 0.441 (11) | 0.487 ( 7) | 1.21 |
| UniTE-src | 0.229 ( 6) | 0.283 ( 9) | 0.232 ( 6) | 0.349 ( 6) | 0.349 (10) | 0.349 ( 6) | 0.444 (10) | 0.479 ( 8) | 0.10 |
| COMETKiwi | 0.230 ( 5) | 0.283 ( 8) | 0.233 ( 5) | 0.349 ( 5) | 0.349 ( 9) | 0.349 ( 5) | 0.444 ( 9) | 0.473 ( 9) | 0.13 |
| GEMBA-GPT-3.5 | 0.208 (10) | 0.301 ( 7) | 0.222 ( 9) | 0.058 (14) | 0.426 ( 6) | 0.316 (10) | 0.452 ( 8) | 0.461 (10) | 5.00 |
| COMET-QE | 0.225 ( 8) | 0.277 (11) | 0.228 ( 8) | 0.342 ( 7) | 0.342 (12) | 0.342 ( 8) | 0.441 (12) | 0.457 (11) | 0.01 |
| MaTESe-QE | 0.119 (16) | 0.242 (13) | 0.129 (15) | -0.391 (17) | 0.422 ( 7) | 0.181 (16) | 0.457 ( 7) | 0.456 (12) | 0.00 |
| MS-COMET-QE-22 | 0.184 (13) | 0.226 (15) | 0.186 (13) | 0.279 (11) | 0.279 (15) | 0.279 (13) | 0.421 (15) | 0.456 (13) | 1.46 |
| SEScore | 0.211 ( 9) | 0.261 (12) | 0.214 (10) | 0.322 ( 9) | 0.322 (13) | 0.322 ( 9) | 0.435 (13) | 0.452 (14) | 0.39 |
| MEE4 | 0.191 (12) | 0.236 (14) | 0.194 (12) | 0.290 (10) | 0.291 (14) | 0.290 (12) | 0.425 (14) | 0.429 (15) | 0.01 |
| HWTSC-Teacher-Sim | 0.122 (15) | 0.150 (16) | 0.123 (16) | 0.185 (12) | 0.185 (16) | 0.185 (15) | 0.390 (16) | 0.403 (16) | 0.15 |
| REUSE | 0.046 (17) | 0.057 (17) | 0.047 (17) | 0.070 (13) | 0.070 (17) | 0.070 (17) | 0.352 (17) | 0.354 (17) | 0.01 |
| Constant-Metric | 0.000 (18) | 0.000 (18) | 0.000 (18) | -1.000 (18) | 0.000 (18) | 0.000 (18) | 0.342 (18) | 0.339 (18) | 0.00 |

Table 12: The correlations (and metric ranks) for the no-grouping correlation on the WMT'22 en-de dataset.

| Metric | $\tau_a$ | $\tau_b$ | $\tau_c$ | $\tau_{10}$ | $\tau_{13}$ | $\tau_{14}$ | $acc_{eq}$ | $acc^*_{eq}$ | $\epsilon^*$ |
|---|---|---|---|---|---|---|---|---|---|
| Metric-X | 0.174 ( 1) | 0.270 ( 4) | 0.269 ( 1) | 0.381 ( 1) | 0.385 ( 3) | 0.384 ( 1) | 0.325 (11) | 0.605 ( 1) | 0.04 |
| UniTE | 0.162 ( 3) | 0.278 ( 3) | 0.255 ( 2) | 0.322 ( 3) | 0.382 ( 4) | 0.368 ( 3) | 0.425 ( 6) | 0.595 ( 2) | 0.14 |
| COMET-22 | 0.163 ( 2) | 0.258 ( 5) | 0.255 ( 3) | 0.366 ( 2) | 0.371 ( 5) | 0.370 ( 2) | 0.325 (12) | 0.594 ( 3) | 0.11 |
| MaTESe | 0.080 (12) | 0.281 ( 2) | 0.207 ( 6) | -0.459 (16) | 0.391 ( 2) | 0.171 (13) | 0.582 ( 1) | 0.582 ( 4) | 0.00 |
| UniTE-src | 0.123 ( 5) | 0.205 ( 9) | 0.190 ( 7) | 0.221 ( 8) | 0.277 ( 9) | 0.266 ( 6) | 0.406 ( 8) | 0.582 ( 5) | 0.12 |
| GEMBA-GPT-4 | 0.106 ( 9) | 0.322 ( 1) | 0.241 ( 4) | -0.367 (15) | 0.487 ( 1) | 0.237 (10) | 0.567 ( 3) | 0.573 ( 6) | 4.00 |
| MaTESe-QE | 0.059 (15) | 0.234 ( 7) | 0.175 (10) | -0.573 (17) | 0.320 ( 8) | 0.121 (15) | 0.572 ( 2) | 0.572 ( 7) | 0.00 |
| COMETKiwi | 0.116 ( 6) | 0.181 (12) | 0.178 ( 9) | 0.254 ( 6) | 0.259 (11) | 0.259 ( 7) | 0.301 (14) | 0.572 ( 8) | 0.16 |
| BLEURT-20 | 0.149 ( 4) | 0.254 ( 6) | 0.233 ( 5) | 0.289 ( 4) | 0.344 ( 6) | 0.334 ( 4) | 0.419 ( 7) | 0.568 ( 9) | 0.09 |
| MS-COMET-22 | 0.107 ( 8) | 0.169 (13) | 0.166 (11) | 0.241 ( 7) | 0.244 (13) | 0.244 ( 9) | 0.294 (15) | 0.565 (10) | 4.65 |
| COMET-QE | 0.094 (11) | 0.138 (14) | 0.144 (14) | 0.179 (10) | 0.182 (14) | 0.182 (12) | 0.285 (17) | 0.555 (11) | 0.01 |
| SEScore | 0.114 ( 7) | 0.182 (10) | 0.180 ( 8) | 0.269 ( 5) | 0.270 (10) | 0.270 ( 5) | 0.291 (16) | 0.554 (12) | 1.30 |
| MS-COMET-QE-22 | 0.051 (16) | 0.080 (16) | 0.076 (16) | 0.116 (12) | 0.118 (16) | 0.118 (16) | 0.264 (18) | 0.550 (13) | 6.50 |
| HWTSC-Teacher-Sim | 0.067 (14) | 0.106 (15) | 0.103 (15) | 0.123 (11) | 0.149 (15) | 0.147 (14) | 0.328 (10) | 0.545 (14) | 0.34 |
| GEMBA-GPT-3.5 | 0.078 (13) | 0.209 ( 8) | 0.151 (13) | -0.344 (14) | 0.324 ( 7) | 0.189 (11) | 0.509 ( 5) | 0.545 (15) | 15.00 |
| MEE4 | 0.100 (10) | 0.182 (11) | 0.157 (12) | 0.201 ( 9) | 0.252 (12) | 0.246 ( 8) | 0.394 ( 9) | 0.539 (16) | 0.13 |
| REUSE | -0.052 (18) | -0.074 (18) | -0.081 (18) | -0.134 (13) | -0.093 (18) | -0.087 (18) | 0.319 (13) | 0.534 (17) | 0.47 |
| Constant-Metric | 0.000 (17) | 0.000 (17) | 0.000 (17) | -1.000 (18) | 0.000 (17) | 0.000 (17) | 0.534 ( 4) | 0.534 (18) | 0.00 |

Table 13: The correlations (and metric ranks) for the group-by-item correlation on the WMT'22 en-de dataset.

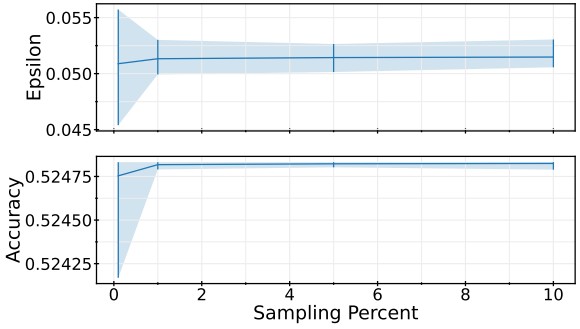

we argue that the results presented in this work apply to either normalization technique, but larger changes will likely be observed under using the Google method due to the increase in number of ties.

Figure 7: The plots show the best $\epsilon^*$ found to maximize $acc_{eq}$ when the $\binom{n}{2}$ pairs are downsampled (percent shown on $x$-axis). The $\epsilon^*$ is then used to calculate $acc_{eq}$ on all $\binom{n}{2}$ pairs. Each sampling rate was run 30 times and the maximum, minimum, and mean values are shown in the plots. Even with as little as 10% of the possible pairs, the approximation is very good.

| Metric | $\tau_a$ | $\tau_b$ | $\tau_c$ | $\tau_{10}$ | $\tau_{13}$ | $\tau_{14}$ | $\text{acc}_{eq}$ | $\text{acc}^*_{eq}$ | $\epsilon^*$ |
|---|---|---|---|---|---|---|---|---|---|
| Metric-X | 0.285 ( 3) | 0.351 ( 3) | 0.293 ( 3) | 0.434 ( 3) | 0.434 ( 5) | 0.434 ( 3) | 0.469 ( 5) | 0.524 ( 1) | 0.05 |
| COMET-22 | 0.290 ( 1) | 0.357 ( 1) | 0.299 ( 1) | 0.441 ( 1) | 0.441 ( 3) | 0.441 ( 1) | 0.472 ( 3) | 0.515 ( 2) | 0.16 |
| UniTE | 0.286 ( 2) | 0.352 ( 2) | 0.294 ( 2) | 0.434 ( 2) | 0.434 ( 4) | 0.434 ( 2) | 0.470 ( 4) | 0.511 ( 3) | 0.14 |
| MaTESe | 0.167 (14) | 0.314 ( 6) | 0.183 (14) | -0.255 (16) | 0.508 ( 2) | 0.253 (14) | 0.499 ( 1) | 0.494 ( 4) | 0.00 |
| GEMBA-GPT-4 | 0.194 (11) | 0.338 ( 4) | 0.213 (11) | -0.168 (15) | 0.542 ( 1) | 0.293 (11) | 0.480 ( 2) | 0.486 ( 5) | 4.00 |
| BLEURT-20 | 0.272 ( 4) | 0.335 ( 5) | 0.280 ( 4) | 0.413 ( 4) | 0.414 ( 8) | 0.414 ( 4) | 0.463 ( 6) | 0.486 ( 6) | 0.04 |
| MS-COMET-22 | 0.223 ( 8) | 0.274 (11) | 0.230 ( 9) | 0.338 ( 8) | 0.338 (12) | 0.338 (12) | 0.439 (12) | 0.482 ( 7) | 1.30 |
| UniTE-src | 0.229 ( 6) | 0.282 ( 9) | 0.235 ( 6) | 0.347 ( 6) | 0.347 (10) | 0.347 ( 6) | 0.441 (10) | 0.473 ( 8) | 0.09 |
| COMETKiwi | 0.229 ( 5) | 0.282 ( 8) | 0.236 ( 5) | 0.347 ( 5) | 0.347 ( 9) | 0.347 ( 5) | 0.442 ( 9) | 0.471 ( 9) | 0.12 |
| COMET-QE | 0.225 ( 7) | 0.276 (10) | 0.231 ( 7) | 0.341 ( 7) | 0.341 (11) | 0.341 ( 7) | 0.439 (11) | 0.459 (10) | 0.02 |
| MaTESe-QE | 0.117 (16) | 0.236 (13) | 0.131 (15) | -0.401 (17) | 0.415 ( 7) | 0.177 (16) | 0.457 ( 7) | 0.456 (11) | 0.00 |
| GEMBA-GPT-3.5 | 0.207 (10) | 0.299 ( 7) | 0.231 ( 8) | 0.056 (14) | 0.424 ( 6) | 0.315 (10) | 0.451 ( 8) | 0.453 (12) | 5.00 |
| SEScore | 0.209 ( 9) | 0.258 (12) | 0.215 (10) | 0.318 ( 9) | 0.318 (13) | 0.318 ( 9) | 0.432 (13) | 0.448 (13) | 0.43 |
| MS-COMET-QE-22 | 0.185 (13) | 0.228 (15) | 0.190 (13) | 0.280 (11) | 0.280 (15) | 0.280 (13) | 0.419 (15) | 0.445 (14) | 1.33 |
| MEE4 | 0.190 (12) | 0.234 (14) | 0.195 (12) | 0.288 (10) | 0.288 (14) | 0.288 (12) | 0.423 (14) | 0.432 (15) | 0.01 |
| HWTSC-Teacher-Sim | 0.121 (15) | 0.149 (16) | 0.125 (16) | 0.184 (12) | 0.184 (16) | 0.184 (15) | 0.388 (16) | 0.405 (16) | 0.15 |
| REUSE | 0.057 (17) | 0.070 (17) | 0.058 (17) | 0.086 (13) | 0.086 (17) | 0.086 (17) | 0.355 (17) | 0.356 (17) | 0.00 |
| Constant-Metric | 0.000 (18) | 0.000 (18) | 0.000 (18) | -1.000 (18) | 0.000 (18) | 0.000 (18) | 0.346 (18) | 0.343 (18) | 0.00 |

Table 14: The correlations (and metric ranks) for the group-by-system correlation on the WMT'22 en-de dataset.

| Metric | $\tau_a$ | $\tau_b$ | $\tau_c$ | $\tau_{10}$ | $\tau_{13}$ | $\tau_{14}$ | $\text{acc}_{eq}$ | $\text{acc}^*_{eq}$ | $\epsilon^*$ |
|---|---|---|---|---|---|---|---|---|---|
| Metric-X | 0.370 ( 1) | 0.420 ( 1) | 0.372 ( 1) | 0.477 ( 1) | 0.477 ( 4) | 0.477 ( 1) | 0.573 ( 1) | 0.579 ( 1) | 0.02 |
| COMET-22 | 0.352 ( 2) | 0.400 ( 2) | 0.354 ( 2) | 0.454 ( 2) | 0.454 ( 5) | 0.454 ( 2) | 0.564 ( 2) | 0.563 ( 2) | 0.01 |
| UniTE | 0.329 ( 3) | 0.374 ( 3) | 0.331 ( 3) | 0.425 ( 3) | 0.425 ( 6) | 0.425 ( 3) | 0.553 ( 3) | 0.553 ( 3) | 0.03 |
| COMETKiwi | 0.316 ( 4) | 0.359 ( 4) | 0.318 ( 4) | 0.408 ( 4) | 0.408 ( 8) | 0.408 ( 4) | 0.546 ( 5) | 0.548 ( 4) | 0.02 |
| BLEURT-20 | 0.316 ( 5) | 0.359 ( 5) | 0.318 ( 5) | 0.408 ( 5) | 0.408 ( 9) | 0.408 ( 5) | 0.546 ( 4) | 0.542 ( 5) | 0.00 |
| MS-COMET-22 | 0.309 ( 6) | 0.351 ( 7) | 0.311 ( 6) | 0.399 ( 6) | 0.399 (10) | 0.399 ( 6) | 0.542 ( 6) | 0.539 ( 6) | 0.18 |
| UniTE-src | 0.301 ( 7) | 0.342 ( 8) | 0.303 ( 7) | 0.388 ( 7) | 0.388 (11) | 0.388 ( 7) | 0.538 ( 7) | 0.536 ( 7) | 0.01 |
| COMET-QE | 0.300 ( 8) | 0.341 ( 9) | 0.302 ( 8) | 0.387 ( 8) | 0.387 (12) | 0.387 ( 8) | 0.538 ( 8) | 0.536 ( 8) | 0.00 |
| MS-COMET-QE-22 | 0.269 ( 9) | 0.305 (11) | 0.271 (10) | 0.347 ( 9) | 0.347 (13) | 0.347 ( 9) | 0.522 ( 9) | 0.519 ( 9) | 0.00 |
| GEMBA-GPT-3.5 | 0.259 (10) | 0.332 (10) | 0.279 ( 9) | 0.125 (12) | 0.422 ( 7) | 0.334 (10) | 0.488 (10) | 0.484 (10) | 0.00 |
| GEMBA-GPT-4 | 0.245 (11) | 0.358 ( 6) | 0.262 (11) | -0.046 (14) | 0.496 ( 2) | 0.316 (11) | 0.483 (11) | 0.483 (11) | 2.00 |
| MEE4 | 0.185 (12) | 0.210 (14) | 0.186 (12) | 0.238 (10) | 0.239 (14) | 0.239 (12) | 0.481 (12) | 0.477 (12) | 0.00 |
| HWTSC-Teacher-Sim | 0.126 (14) | 0.143 (15) | 0.127 (14) | 0.163 (11) | 0.163 (15) | 0.163 (14) | 0.451 (13) | 0.448 (13) | 0.00 |
| REUSE | 0.069 (16) | 0.078 (16) | 0.069 (16) | 0.088 (13) | 0.088 (16) | 0.088 (16) | 0.422 (14) | 0.418 (14) | 0.00 |
| MaTESe | 0.128 (13) | 0.279 (12) | 0.140 (13) | -0.523 (15) | 0.529 ( 1) | 0.165 (13) | 0.380 (15) | 0.389 (15) | 0.00 |
| MaTESe-QE | 0.093 (15) | 0.229 (13) | 0.103 (15) | -0.636 (16) | 0.493 ( 3) | 0.120 (15) | 0.341 (16) | 0.349 (16) | 0.00 |
| Constant-Metric | 0.000 (17) | 0.000 (17) | 0.000 (17) | -1.000 (17) | 0.000 (17) | 0.000 (17) | 0.225 (17) | 0.230 (17) | 0.00 |

Table 15: The correlations (and metric ranks) for the no-grouping correlation on the WMT'22 en-ru dataset.

| Metric | $\tau_a$ | $\tau_b$ | $\tau_c$ | $\tau_{10}$ | $\tau_{13}$ | $\tau_{14}$ | $\text{acc}_{eq}$ | $\text{acc}^*_{eq}$ | $\epsilon^*$ |
|---|---|---|---|---|---|---|---|---|---|
| Metric-X | 0.239 ( 1) | 0.329 ( 2) | 0.323 ( 1) | 0.444 ( 1) | 0.445 ( 3) | 0.445 ( 1) | 0.402 (10) | 0.606 ( 1) | 0.03 |
| COMET-22 | 0.230 ( 2) | 0.315 ( 3) | 0.309 ( 2) | 0.420 ( 2) | 0.421 ( 4) | 0.421 ( 2) | 0.396 (11) | 0.577 ( 2) | 0.07 |
| UniTE | 0.220 ( 3) | 0.311 ( 4) | 0.297 ( 3) | 0.399 ( 3) | 0.415 ( 5) | 0.412 ( 3) | 0.445 ( 8) | 0.572 ( 3) | 0.05 |
| COMETKiwi | 0.181 ( 5) | 0.247 ( 8) | 0.242 ( 7) | 0.331 ( 5) | 0.332 ( 9) | 0.332 ( 6) | 0.372 (13) | 0.565 ( 4) | 0.05 |
| UniTE-src | 0.180 ( 6) | 0.267 ( 7) | 0.245 ( 6) | 0.322 ( 7) | 0.347 ( 8) | 0.343 ( 5) | 0.463 ( 6) | 0.554 ( 5) | 0.07 |
| GEMBA-GPT-4 | 0.155 ( 8) | 0.356 ( 1) | 0.273 ( 4) | -0.203 (13) | 0.519 ( 1) | 0.290 ( 8) | 0.548 ( 1) | 0.550 ( 6) | 4.00 |
| MS-COMET-22 | 0.177 ( 7) | 0.243 ( 9) | 0.238 ( 8) | 0.327 ( 6) | 0.328 (10) | 0.328 ( 7) | 0.367 (14) | 0.547 ( 7) | 2.47 |
| BLEURT-20 | 0.202 ( 4) | 0.291 ( 6) | 0.269 ( 5) | 0.347 ( 4) | 0.369 ( 7) | 0.367 ( 4) | 0.474 ( 5) | 0.540 ( 8) | 0.05 |
| COMET-QE | 0.148 (10) | 0.200 (13) | 0.197 (11) | 0.262 ( 9) | 0.262 (13) | 0.262 (10) | 0.352 (15) | 0.534 ( 9) | 0.01 |
| MS-COMET-QE-22 | 0.131 (11) | 0.180 (14) | 0.177 (12) | 0.243 (10) | 0.243 (14) | 0.243 (11) | 0.344 (16) | 0.528 (10) | 3.14 |
| MaTESe | 0.075 (14) | 0.293 ( 5) | 0.224 ( 9) | -0.630 (15) | 0.472 ( 2) | 0.126 (14) | 0.520 ( 2) | 0.520 (11) | 0.00 |
| MaTESe-QE | 0.048 (15) | 0.238 (10) | 0.168 (13) | -0.728 (16) | 0.379 ( 6) | 0.085 (15) | 0.499 ( 3) | 0.499 (12) | 0.00 |
| GEMBA-GPT-3.5 | 0.098 (13) | 0.201 (12) | 0.158 (14) | -0.237 (14) | 0.308 (11) | 0.196 (13) | 0.475 ( 4) | 0.494 (13) | 5.00 |
| HWTSC-Teacher-Sim | 0.098 (12) | 0.147 (15) | 0.137 (15) | 0.185 (11) | 0.199 (15) | 0.198 (12) | 0.387 (12) | 0.481 (14) | 0.05 |
| MEE4 | 0.149 ( 9) | 0.222 (11) | 0.201 (10) | 0.267 ( 8) | 0.289 (12) | 0.287 ( 9) | 0.448 ( 7) | 0.473 (15) | 0.03 |
| REUSE | -0.076 (17) | -0.090 (17) | -0.097 (17) | -0.119 (12) | -0.099 (17) | -0.097 (17) | 0.335 (17) | 0.446 (16) | 0.14 |
| Constant-Metric | 0.000 (16) | 0.000 (16) | 0.000 (16) | -1.000 (17) | 0.000 (16) | 0.000 (16) | 0.444 ( 9) | 0.444 (17) | 0.00 |

Table 16: The correlations (and metric ranks) for the group-by-item correlation on the WMT'22 en-ru dataset.

| Metric | $\tau_a$ | $\tau_b$ | $\tau_c$ | $\tau_{10}$ | $\tau_{13}$ | $\tau_{14}$ | $\text{acc}_{eq}$ | $\text{acc}_{eq}^*$ | $\epsilon^*$ |
|---|---|---|---|---|---|---|---|---|---|
| Metric-X | 0.352 ( 1) | 0.401 ( 1) | 0.358 ( 1) | 0.457 ( 1) | 0.457 ( 4) | 0.457 ( 1) | 0.559 ( 1) | 0.569 ( 1) | 0.02 |
| COMET-22 | 0.337 ( 2) | 0.384 ( 2) | 0.343 ( 2) | 0.438 ( 2) | 0.438 ( 5) | 0.438 ( 2) | 0.552 ( 2) | 0.548 ( 2) | 0.03 |
| UniTE | 0.316 ( 3) | 0.360 ( 3) | 0.321 ( 3) | 0.410 ( 3) | 0.410 ( 6) | 0.410 ( 3) | 0.541 ( 3) | 0.543 ( 3) | 0.04 |
| COMETKiwi | 0.308 ( 4) | 0.350 ( 4) | 0.313 ( 4) | 0.399 ( 4) | 0.399 ( 8) | 0.399 ( 4) | 0.537 ( 4) | 0.538 ( 4) | 0.02 |
| COMET-QE | 0.299 ( 6) | 0.340 ( 6) | 0.304 ( 6) | 0.387 ( 6) | 0.387 (10) | 0.387 ( 6) | 0.533 ( 6) | 0.529 ( 5) | 0.00 |
| BLEURT-20 | 0.301 ( 5) | 0.343 ( 5) | 0.306 ( 5) | 0.391 ( 5) | 0.391 ( 9) | 0.391 ( 5) | 0.534 ( 5) | 0.527 ( 6) | 0.00 |
| UniTE-src | 0.291 ( 8) | 0.332 ( 8) | 0.296 ( 8) | 0.378 ( 8) | 0.378 (12) | 0.378 ( 8) | 0.529 ( 8) | 0.526 ( 7) | 0.02 |
| MS-COMET-22 | 0.296 ( 7) | 0.337 ( 7) | 0.301 ( 7) | 0.384 ( 7) | 0.384 (11) | 0.384 ( 7) | 0.531 ( 7) | 0.524 ( 8) | 1.00 |
| MS-COMET-QE-22 | 0.261 ( 9) | 0.297 (11) | 0.266 (10) | 0.337 ( 9) | 0.337 (13) | 0.337 ( 9) | 0.514 ( 9) | 0.505 ( 9) | 0.00 |
| GEMBA-GPT-3.5 | 0.250 (10) | 0.321 (10) | 0.272 ( 9) | 0.113 (12) | 0.410 ( 7) | 0.324 (10) | 0.482 (10) | 0.475 (10) | 0.00 |
| GEMBA-GPT-4 | 0.224 (11) | 0.327 ( 9) | 0.244 (11) | -0.089 (14) | 0.460 ( 3) | 0.288 (11) | 0.471 (11) | 0.473 (11) | 2.00 |
| MEE4 | 0.171 (12) | 0.195 (14) | 0.174 (12) | 0.222 (10) | 0.223 (14) | 0.223 (12) | 0.470 (12) | 0.461 (12) | 0.00 |
| HWTSC-Teacher-Sim | 0.123 (13) | 0.139 (15) | 0.125 (14) | 0.159 (11) | 0.159 (15) | 0.159 (13) | 0.445 (13) | 0.439 (13) | 0.00 |
| REUSE | 0.084 (16) | 0.095 (16) | 0.085 (16) | 0.108 (13) | 0.108 (16) | 0.108 (16) | 0.425 (14) | 0.422 (14) | 0.00 |
| MaTESe | 0.120 (14) | 0.258 (12) | 0.139 (13) | -0.544 (15) | 0.504 ( 1) | 0.153 (14) | 0.381 (15) | 0.387 (15) | 0.00 |
| MaTESe-QE | 0.089 (15) | 0.211 (13) | 0.105 (15) | -0.650 (16) | 0.466 ( 2) | 0.113 (15) | 0.345 (16) | 0.353 (16) | 0.00 |
| Constant-Metric | 0.000 (17) | 0.000 (17) | 0.000 (17) | -1.000 (17) | 0.000 (17) | 0.000 (17) | 0.233 (17) | 0.242 (17) | 0.00 |

Table 17: The correlations (and metric ranks) for the group-by-system correlation on the WMT'22 en-ru dataset.

| Metric | $\tau_a$ | $\tau_b$ | $\tau_c$ | $\tau_{10}$ | $\tau_{13}$ | $\tau_{14}$ | $\text{acc}_{eq}$ | $\text{acc}_{eq}^*$ | $\epsilon^*$ |
|---|---|---|---|---|---|---|---|---|---|
| Metric-X | 0.362 ( 1) | 0.421 ( 1) | 0.364 ( 1) | 0.489 ( 1) | 0.489 ( 1) | 0.489 ( 1) | 0.551 ( 1) | 0.565 ( 1) | 0.06 |
| COMET-22 | 0.361 ( 2) | 0.420 ( 2) | 0.363 ( 2) | 0.488 ( 2) | 0.488 ( 2) | 0.488 ( 2) | 0.551 ( 2) | 0.555 ( 2) | 0.14 |
| MaTESe | 0.295 ( 7) | 0.382 ( 3) | 0.307 ( 4) | 0.244 (12) | 0.471 ( 5) | 0.399 ( 7) | 0.541 ( 3) | 0.536 ( 3) | 0.00 |
| COMET-QE | 0.306 ( 3) | 0.356 ( 6) | 0.308 ( 3) | 0.414 ( 3) | 0.414 ( 6) | 0.414 ( 3) | 0.523 ( 4) | 0.520 ( 4) | 0.00 |
| COMETKiwi | 0.303 ( 6) | 0.352 ( 9) | 0.304 ( 7) | 0.409 ( 6) | 0.409 (10) | 0.409 ( 6) | 0.522 ( 7) | 0.520 ( 5) | 0.03 |
| BLEURT-20 | 0.303 ( 5) | 0.352 ( 8) | 0.305 ( 6) | 0.410 ( 5) | 0.410 ( 9) | 0.410 ( 5) | 0.522 ( 6) | 0.517 ( 6) | 0.00 |
| UniTE | 0.305 ( 4) | 0.354 ( 7) | 0.306 ( 5) | 0.412 ( 4) | 0.412 ( 7) | 0.412 ( 4) | 0.523 ( 5) | 0.516 ( 7) | 0.05 |
| GEMBA-GPT-4 | 0.268 (11) | 0.370 ( 4) | 0.284 (10) | 0.108 (16) | 0.486 ( 4) | 0.362 (11) | 0.513 (11) | 0.513 ( 8) | 4.00 |
| MS-COMET-22 | 0.288 ( 8) | 0.335 (10) | 0.289 ( 8) | 0.389 ( 7) | 0.389 (11) | 0.389 ( 8) | 0.514 ( 8) | 0.510 ( 9) | 0.02 |
| UniTE-src | 0.286 ( 9) | 0.332 (11) | 0.287 ( 9) | 0.386 ( 8) | 0.386 (12) | 0.386 ( 9) | 0.513 (10) | 0.508 (10) | 0.00 |
| SEScore | 0.279 (10) | 0.324 (13) | 0.280 (11) | 0.377 ( 9) | 0.377 (13) | 0.377 (10) | 0.510 (12) | 0.506 (11) | 0.00 |
| MaTESe-QE | 0.251 (13) | 0.328 (12) | 0.261 (13) | 0.164 (14) | 0.411 ( 8) | 0.339 (13) | 0.513 ( 9) | 0.506 (12) | 0.00 |
| GEMBA-GPT-3.5 | 0.254 (12) | 0.360 ( 5) | 0.273 (12) | 0.047 (17) | 0.486 ( 3) | 0.343 (12) | 0.499 (13) | 0.499 (13) | 0.00 |
| MS-COMET-QE-22 | 0.238 (14) | 0.277 (14) | 0.239 (14) | 0.322 (10) | 0.322 (14) | 0.322 (14) | 0.489 (14) | 0.486 (14) | 0.00 |
| HWTSC-Teacher-Sim | 0.227 (15) | 0.264 (15) | 0.228 (15) | 0.307 (11) | 0.307 (15) | 0.307 (15) | 0.484 (15) | 0.477 (15) | 0.00 |
| MEE4 | 0.163 (16) | 0.189 (16) | 0.164 (16) | 0.220 (13) | 0.220 (16) | 0.220 (16) | 0.452 (16) | 0.449 (16) | 0.00 |
| REUSE | 0.100 (17) | 0.116 (17) | 0.101 (17) | 0.135 (15) | 0.135 (17) | 0.135 (17) | 0.420 (17) | 0.417 (17) | 0.00 |
| Constant-Metric | 0.000 (18) | 0.000 (18) | 0.000 (18) | -1.000 (18) | 0.000 (18) | 0.000 (18) | 0.260 (18) | 0.267 (18) | 0.00 |

Table 18: The correlations (and metric ranks) for the no-grouping correlation on the WMT'22 zh-en dataset.

| Metric | $\tau_a$ | $\tau_b$ | $\tau_c$ | $\tau_{10}$ | $\tau_{13}$ | $\tau_{14}$ | $\text{acc}_{eq}$ | $\text{acc}_{eq}^*$ | $\epsilon^*$ |
|---|---|---|---|---|---|---|---|---|---|
| Metric-X | 0.191 ( 3) | 0.255 ( 5) | 0.245 ( 4) | 0.343 ( 2) | 0.344 ( 5) | 0.344 ( 3) | 0.389 (10) | 0.544 ( 1) | 0.06 |
| COMET-22 | 0.198 ( 1) | 0.266 ( 2) | 0.255 ( 1) | 0.361 ( 1) | 0.361 ( 3) | 0.361 ( 1) | 0.392 ( 9) | 0.536 ( 2) | 0.15 |
| GEMBA-GPT-4 | 0.175 ( 4) | 0.321 ( 1) | 0.252 ( 2) | -0.071 (13) | 0.450 ( 1) | 0.314 ( 4) | 0.518 ( 1) | 0.527 ( 3) | 4.00 |
| UniTE | 0.191 ( 2) | 0.261 ( 3) | 0.246 ( 3) | 0.335 ( 3) | 0.350 ( 4) | 0.347 ( 2) | 0.420 ( 5) | 0.516 ( 4) | 0.29 |
| MaTESe | 0.127 (10) | 0.225 ( 7) | 0.180 (10) | -0.108 (14) | 0.289 ( 8) | 0.220 (11) | 0.498 ( 2) | 0.512 ( 5) | 1.00 |
| COMETKiwi | 0.159 ( 6) | 0.213 ( 9) | 0.204 ( 6) | 0.288 ( 6) | 0.289 ( 9) | 0.289 ( 7) | 0.372 (11) | 0.509 ( 6) | 0.16 |
| UniTE-src | 0.164 ( 5) | 0.226 ( 6) | 0.211 ( 5) | 0.293 ( 5) | 0.306 ( 6) | 0.304 ( 5) | 0.407 ( 7) | 0.508 ( 7) | 0.24 |
| GEMBA-GPT-3.5 | 0.123 (12) | 0.256 ( 4) | 0.199 ( 8) | -0.271 (16) | 0.377 ( 2) | 0.225 (10) | 0.494 ( 3) | 0.495 ( 8) | 5.00 |
| MaTESe-QE | 0.097 (14) | 0.181 (12) | 0.141 (14) | -0.195 (15) | 0.226 (12) | 0.169 (14) | 0.484 ( 4) | 0.494 ( 9) | 1.00 |
| COMET-QE | 0.126 (11) | 0.165 (13) | 0.159 (12) | 0.219 ( 9) | 0.219 (13) | 0.219 (12) | 0.355 (15) | 0.483 (10) | 0.01 |
| MS-COMET-22 | 0.144 ( 8) | 0.196 (10) | 0.187 ( 9) | 0.270 ( 7) | 0.270 (10) | 0.270 ( 8) | 0.365 (13) | 0.483 (11) | 3.49 |
| MS-COMET-QE-22 | 0.117 (13) | 0.158 (14) | 0.150 (13) | 0.214 (10) | 0.214 (14) | 0.214 (13) | 0.351 (16) | 0.479 (12) | 1.99 |
| SEScore | 0.158 ( 7) | 0.214 ( 8) | 0.203 ( 7) | 0.295 ( 4) | 0.295 ( 7) | 0.295 ( 6) | 0.371 (12) | 0.472 (13) | 1.13 |
| HWTSC-Teacher-Sim | 0.086 (15) | 0.116 (15) | 0.110 (15) | 0.148 (11) | 0.156 (15) | 0.155 (15) | 0.357 (14) | 0.440 (14) | 0.14 |
| MEE4 | 0.137 ( 9) | 0.190 (11) | 0.177 (11) | 0.243 ( 8) | 0.256 (11) | 0.254 ( 9) | 0.393 ( 8) | 0.437 (15) | 0.07 |
| REUSE | -0.025 (17) | -0.019 (17) | -0.026 (17) | -0.022 (12) | -0.010 (17) | -0.010 (17) | 0.312 (17) | 0.420 (16) | 0.11 |
| Constant-Metric | 0.000 (16) | 0.000 (16) | 0.000 (16) | -1.000 (17) | 0.000 (16) | 0.000 (16) | 0.416 ( 6) | 0.416 (17) | 0.00 |

Table 19: The correlations (and metric ranks) for the group-by-item correlation on the WMT'22 zh-en dataset.

| Metric | $\tau_a$ | $\tau_b$ | $\tau_c$ | $\tau_{10}$ | $\tau_{13}$ | $\tau_{14}$ | $\text{acc}_{eq}$ | $\text{acc}^*_{eq}$ | $\epsilon^*$ |
|---|---|---|---|---|---|---|---|---|---|
| Metric-X | 0.353 ( 1) | 0.411 ( 1) | 0.358 ( 1) | 0.478 ( 1) | 0.478 ( 1) | 0.478 ( 1) | 0.544 ( 1) | 0.557 ( 1) | 0.05 |
| COMET-22 | 0.350 ( 2) | 0.408 ( 2) | 0.355 ( 2) | 0.475 ( 2) | 0.475 ( 2) | 0.475 ( 2) | 0.543 ( 2) | 0.546 ( 2) | 0.08 |
| MaTESe | 0.288 ( 7) | 0.373 ( 3) | 0.300 ( 4) | 0.231 (12) | 0.462 ( 4) | 0.390 ( 7) | 0.536 ( 3) | 0.530 ( 3) | 0.00 |
| COMET-QE | 0.306 ( 3) | 0.355 ( 4) | 0.310 ( 3) | 0.414 ( 3) | 0.414 ( 6) | 0.414 ( 3) | 0.521 ( 4) | 0.518 ( 4) | 0.00 |
| COMETKiwi | 0.295 ( 4) | 0.343 ( 7) | 0.300 ( 5) | 0.399 ( 4) | 0.399 ( 8) | 0.399 ( 4) | 0.515 ( 5) | 0.516 ( 5) | 0.09 |
| BLEURT-20 | 0.289 ( 6) | 0.336 ( 9) | 0.293 ( 7) | 0.392 ( 6) | 0.392 (10) | 0.392 ( 6) | 0.512 ( 7) | 0.507 ( 6) | 0.00 |
| MaTESe-QE | 0.247 (12) | 0.323 (10) | 0.257 (13) | 0.154 (14) | 0.405 ( 7) | 0.333 (12) | 0.510 ( 8) | 0.506 ( 7) | 0.00 |
| UniTE | 0.290 ( 5) | 0.338 ( 8) | 0.294 ( 6) | 0.393 ( 5) | 0.393 ( 9) | 0.393 ( 5) | 0.513 ( 6) | 0.505 ( 8) | 0.01 |
| GEMBA-GPT-4 | 0.250 (11) | 0.347 ( 5) | 0.269 (11) | 0.070 (16) | 0.461 ( 5) | 0.338 (11) | 0.502 (11) | 0.504 ( 9) | 4.00 |
| MS-COMET-22 | 0.277 ( 8) | 0.322 (11) | 0.281 ( 8) | 0.376 ( 7) | 0.376 (11) | 0.376 ( 8) | 0.506 ( 9) | 0.502 (10) | 0.00 |
| SEScore | 0.268 (10) | 0.311 (13) | 0.272 (10) | 0.362 ( 9) | 0.362 (13) | 0.362 (10) | 0.502 (12) | 0.499 (11) | 0.01 |
| UniTE-src | 0.276 ( 9) | 0.321 (12) | 0.280 ( 9) | 0.374 ( 8) | 0.374 (12) | 0.374 ( 9) | 0.506 (10) | 0.498 (12) | 0.00 |
| GEMBA-GPT-3.5 | 0.241 (13) | 0.345 ( 6) | 0.264 (12) | 0.022 (17) | 0.470 ( 3) | 0.326 (13) | 0.492 (13) | 0.491 (13) | 0.00 |
| MS-COMET-QE-22 | 0.231 (14) | 0.268 (14) | 0.234 (14) | 0.312 (10) | 0.312 (14) | 0.312 (14) | 0.483 (14) | 0.479 (14) | 0.01 |
| HWTSC-Teacher-Sim | 0.228 (15) | 0.265 (15) | 0.231 (15) | 0.308 (11) | 0.308 (15) | 0.308 (15) | 0.482 (15) | 0.475 (15) | 0.00 |
| MEE4 | 0.150 (16) | 0.174 (16) | 0.152 (16) | 0.202 (13) | 0.202 (16) | 0.202 (16) | 0.443 (16) | 0.441 (16) | 0.00 |
| REUSE | 0.108 (17) | 0.125 (17) | 0.109 (17) | 0.146 (15) | 0.146 (17) | 0.146 (17) | 0.422 (17) | 0.419 (17) | 0.00 |
| Constant-Metric | 0.000 (18) | 0.000 (18) | 0.000 (18) | -1.000 (18) | 0.000 (18) | 0.000 (18) | 0.265 (18) | 0.270 (18) | 0.00 |

Table 20: The correlations (and metric ranks) for the group-by-system correlation on the WMT'22 zh-en dataset.

| Metric | $\tau_a$ | $\tau_b$ | $\tau_c$ | $\tau_{10}$ | $\tau_{13}$ | $\tau_{14}$ | $\text{acc}_{eq}$ | $\text{acc}^*_{eq}$ | $\epsilon^*$ |
|---|---|---|---|---|---|---|---|---|---|
| Metric-X | 0.239 ( 1) | 0.312 ( 2) | 0.295 ( 1) | 0.402 ( 1) | 0.403 ( 3) | 0.403 ( 1) | 0.439 ( 8) | 0.594 ( 1) | 0.03 |
| UniTE | 0.224 ( 3) | 0.299 ( 3) | 0.275 ( 3) | 0.366 ( 3) | 0.380 ( 5) | 0.378 ( 3) | 0.483 ( 4) | 0.569 ( 2) | 0.03 |
| COMET-22 | 0.230 ( 2) | 0.298 ( 4) | 0.281 ( 2) | 0.380 ( 2) | 0.381 ( 4) | 0.381 ( 2) | 0.433 (10) | 0.564 ( 3) | 0.05 |
| COMETKiwi | 0.187 ( 5) | 0.242 ( 7) | 0.229 ( 6) | 0.310 ( 4) | 0.311 ( 8) | 0.311 ( 5) | 0.411 (11) | 0.561 ( 4) | 0.03 |
| UniTE-src | 0.187 ( 4) | 0.262 ( 6) | 0.231 ( 5) | 0.302 ( 5) | 0.324 ( 7) | 0.320 ( 4) | 0.504 ( 2) | 0.547 ( 5) | 0.02 |
| MS-COMET-22 | 0.180 ( 6) | 0.234 ( 8) | 0.221 ( 8) | 0.300 ( 6) | 0.301 ( 9) | 0.301 ( 6) | 0.405 (12) | 0.540 ( 6) | 1.14 |
| GEMBA-GPT-4 | 0.156 ( 7) | 0.339 ( 1) | 0.267 ( 4) | -0.237 (12) | 0.480 ( 1) | 0.262 ( 8) | 0.524 ( 1) | 0.525 ( 7) | 4.00 |
| COMET-QE | 0.151 ( 9) | 0.192 (11) | 0.184 (10) | 0.241 ( 8) | 0.241 (12) | 0.241 ( 9) | 0.390 (13) | 0.513 ( 8) | 0.00 |
| MS-COMET-QE-22 | 0.133 (10) | 0.173 (13) | 0.163 (12) | 0.223 ( 9) | 0.224 (13) | 0.223 (10) | 0.382 (14) | 0.512 ( 9) | 1.42 |
| MEE4 | 0.154 ( 8) | 0.216 (10) | 0.189 ( 9) | 0.249 ( 7) | 0.269 (10) | 0.267 ( 7) | 0.487 ( 3) | 0.487 (10) | 0.00 |
| HWTSC-Teacher-Sim | 0.120 (11) | 0.165 (14) | 0.149 (13) | 0.196 (10) | 0.208 (14) | 0.208 (11) | 0.435 ( 9) | 0.480 (11) | 0.02 |
| MaTESe | 0.075 (13) | 0.278 ( 5) | 0.225 ( 7) | -0.646 (14) | 0.433 ( 2) | 0.114 (13) | 0.469 ( 5) | 0.469 (12) | 0.00 |
| GEMBA-GPT-3.5 | 0.093 (12) | 0.179 (12) | 0.146 (14) | -0.264 (13) | 0.261 (11) | 0.171 (12) | 0.456 ( 6) | 0.460 (13) | 5.00 |
| MaTESe-QE | 0.047 (14) | 0.223 ( 9) | 0.165 (11) | -0.741 (15) | 0.354 ( 6) | 0.075 (14) | 0.441 ( 7) | 0.441 (14) | 0.00 |
| REUSE | -0.064 (16) | -0.069 (16) | -0.074 (16) | -0.085 (11) | -0.066 (16) | -0.064 (16) | 0.378 (15) | 0.401 (15) | 0.02 |
| Constant-Metric | 0.000 (15) | 0.000 (15) | 0.000 (15) | -1.000 (16) | 0.000 (15) | 0.000 (15) | 0.371 (16) | 0.371 (16) | 0.00 |

Table 21: The group-by-item segment-level correlations on WMT'22 en-ru using the Unbabel MQM score normalization largely follow the results in the main body of this work, which use the Google normalization.