# OpenReview forum: "Ties Matter: Meta-Evaluating Modern Metrics with Pairwise Accuracy and Tie Calibration"
_EMNLP/2023/Conference — EMNLP 2023 Main_

### Official Review · Reviewer_FkvL · 2023-08-04

**Soundness:** 4

**Excitement:**

4: Strong: This paper deepens the understanding of some phenomenon or lowers the barriers to an existing research direction.

**Paper Topic And Main Contributions:**

This paper addresses MT meta-evaluation by examining the problems caused by ties in Kendall’s τ.
The authors propose a  version of pairwise accuracy that grants metrics for correctly predicting ties.
The authors also propose a tie calibration procedure that automatically introduces ties into the metric scores. This calibration enables a fair comparison between metrics that do and do not predict ties.
The experiments were conducted on the Multidimensional Quality Metrics (MQM) ratings data set of the WMT22 metrics shared task for three language pairs (English-German, Chinese-English and English-Russian).
The experimental results show evidence that the proposed method with tie calibration  lead to fairer ranking-based assessments of metric
performance.

**Questions For The Authors:**

Q1: Can the authors introduce some real examples of the tie problem  which is the core if this work? Indeed, for people who are not very familiar with the tie problem, it is difficult to grasp the idea of the paper.

Q2: As stated by the authors in the Limitation Section,  the tie calibration algorithm uses  absolute differences in metric scores
I was wondering why not testing on relative scores as well? Does it require more computation? I guess it's a "simple" normalization process, so could the authors provide these results as well?

**Reasons To Accept:**

- Overall, the paper is clear and well written
- Interesting problem which includes fairness in metrics comparison
- The paper shows improvements over the baselines

**Reasons To Reject:**

- It is difficult to grasp evidence on "fair" comparison between the different metrics in the paper.
- The importance of the tie problem may be questionable and it is not clear to me whether this has a significant impact on metrics comparison.

**Reproducibility:**

2: Would be hard pressed to reproduce the results. The contribution depends on data that are simply not available outside the author's institution or consortium; not enough details are provided.

**Reviewer Confidence:**

2: Willing to defend my evaluation, but it is fairly likely that I missed some details, didn't understand some central points, or can't be sure about the novelty of the work.

**Typos Grammar Style And Presentation Improvements:**

For readers who are not very familiar with the tie problem, and for a better understanding of the task, the tie problem should be introduced earlier in the paper via some real examples.

Typo:
- Line 591 "also" is doubled

---

> ### Author Rebuttal · Authors · 2023-08-24
>
> Thank you very much for your feedback on our work.
>
> Regarding the evidence on a “fair” comparison: It is true that we cannot prove this comparison is more fair (and we mention this in the Limitations). Our argument for making this claim is that our proposed method addresses bias related to ties, which we do believe results in a more fair comparison.
>
> Regarding the importance of ties and Q1: In Section 5.1 (Figure 2), we include a toy motivating example in which the presence of ties results in an incorrect meta-evaluation of two different metrics with previously proposed taus. In short, incorrectly handling ties can result in erroneous conclusions about which metric is better than another.
>
> Ties are a very important problem for metric comparison because incorrectly handling them could lead to unfair comparisons between metrics. The toy example does happen in practice with new metrics like GEMBA that output a large number of ties (see Table 4). Not handling ties will lead to wrong conclusions about these LLM-based metrics. Further, not handling ties opens the door for a metric to cheat the evaluation. In Section 5.2, we introduce the NaN problem, which is a direct result of incorrectly handling ties. We show how a metric could artificially inject ties in order to introduce NaN correlations and increase its performance overall. A metric could have used the NaN problem in the Workshop on Machine Translation’s Metrics Shared Task to improve its overall performance, effectively cheating the evaluation. Correctly handling ties is necessary for making decisions about which metrics agree more frequently with humans than others.
>
> Regarding Q2: The computational cost of using absolute or relative differences for tie calibration is the same. The main difference between them is whether you assume absolute or relative differences are comparable across the full range of scores that can be predicted by a metric. Ideally, metric developers would specify which is true for their metric, and tie calibration would use the corresponding difference. Since the type of difference used is not critical to our work, we will include results about relative differences in the appendix in the updated version.

---

### Official Review · Reviewer_HC2C · 2023-08-05

**Soundness:** 4

**Excitement:**

5: Transformative: This paper is likely to change its subfield or computational linguistics broadly. It should be considered for a best paper award. This paper changes the current understanding of some phenomenon, shows a widely held practice to be erroneous in someway, enables a promising direction of research for a (broad or narrow) topic, or creates an exciting new technique.

**Paper Topic And Main Contributions:**

In this paper, the authors address how existing metrics for the meta-evaluation of machine translation (MT) metrics neglect to handle or ineffectively process ties in pairwise evaluations. Using a motivating example, they show that  Kendall’s Tau, the widely used rank-based correlation statistic for MT meta-evaluation, and its variants unfairly penalize metrics for predicting ties. They also show that Kendall’s Tau can be artificially inflated by introducing ties for challenging groups, resulting in the challenging groups being excluded from meta-evaluation. Finally, the authors propose a new ranking-based meta-evaluation metric—pairwise accuracy with ties—and a method (tie calibration) that allows pairwise accuracy to be fairly calculated for metrics that rarely predict ties.

**Questions For The Authors:**

Question A: Is there a downside to using Kendall’s Tau (given the weaknesses you mention in the paper) as the rank-based correlation statistic for tie calibration?

**Reasons To Accept:**

This paper points out a major flaw with a nearly ubiquitous meta-evaluation metric. The authors included straightforward, and therefore very convincing, motivating examples that make it the use of Kendall’s Tau almost seem silly now. They propose a simple alternative meta-evaluation metric (pairwise accuracy) and include a way to apply this metric to MT metrics that rarely assign ties (tie calibration), making it easy for the MT community to adopt this meta-evaluation metric almost immediately. Additionally, the epsilon “tie calibration” value makes strides towards the interpretability of MT metrics.

**Reasons To Reject:**

This is hardly a reason to reject, but if I HAVE to say something: I think BLEURT should be included in Table 6 since it’s mentioned in Section 7 and you did analysis using BLEURT in 8.2.

**Reproducibility:**

5: Could easily reproduce the results.

**Reviewer Confidence:**

4: Quite sure. I tried to check the important points carefully. It's unlikely, though conceivable, that I missed something that should affect my ratings.

**Typos Grammar Style And Presentation Improvements:**

Section 4 could be moved earlier in the paper since it is motivation for the work and not technical.

---

> ### Author Rebuttal · Authors · 2023-08-24
>
> Thanks so much for your review of our work.
>
> Regarding your suggestion about including BLEURT: We think that is a good suggestion and we will incorporate it into an updated version. We did not include BLEURT in Table 6 because we only considered the primary submissions to the WMT’22 metrics shared task, which does not include BLEURT. We had to use BLEURT in our analysis in 8.2 because we evaluate the metric on WMT’21 and WMT’22, and some of the shared task submissions may have used the WMT’21 data for training or development, which could be a confounding factor in our analysis.
>
> Regarding your comment about using tau for tie calibration: Do you mean a tau that does not correctly handle ties or our proposal, tau_eq, that does?
>
> If your question is about a tau that does not correctly handle ties, then yes, there is a problem with using that tau with tie calibration. For example, tau_13 does not include ties in its formula. Tie calibration could introduce a tie in order to convert from a discordant pair to a tie, thereby removing it from the formula, resulting in a higher tau. This is clearly not right because the new tie is wrong according to the human scores, but the tau value increases.
>
> If your question is about tau_eq, then no, there is no disadvantage. The same epsilon value will be found if you run tie calibration with tau_eq and acc_eq. The only difference in the interpretation of the agreement statistic (accuracy between 0 and 1 versus a tau between -1 and 1).
>
> We will include a discussion of this in the new version of the paper.

---

### Official Review · Reviewer_7hqG · 2023-08-05

**Soundness:** 5

**Excitement:**

4: Strong: This paper deepens the understanding of some phenomenon or lowers the barriers to an existing research direction.

**Paper Topic And Main Contributions:**

 This paper addresses a blind spot in meta evaluation of machine translation metrics.
Currently, meta evaluations (e.g., in WMT Metrics Shared Task) are based on correlation statistics that have a flaw in handling ties assessment scores. This issue is increasingly problematic as generation quality is approaching near human, and some metrics based on LLMs (such as chatgpt) have appear to top the charts.

Main contributions:
 #1) Exposes a blind spot: a) shows how ties can affect correlations and lead to gaming of meta-evaluation.
 #2) Fixes the blind spot: Proposes a meta evaluation metric that is intuitive and fixes the ties issue of #1. Currently, correctly predicting ties lead to no reward in meta-evaluation, however, as described in this paper, instead the correct prediction of ties should be rewarded.
 #3) Sometimes correctly predicting the ties are hard in practice for some auto evaluation metrics. Eg. many recent metrics are designed to output real/float numbers which are not often unequal in the purest sense. This paper introduces a calibration procedure that treats small differences in float numbers as tie, which could lead to increased performance


**Questions For The Authors:**

In settings where there are no ties, or at least not as many as the WMT22 MQM languages, how does your proposed metric perform.
We are interested to know, would your metric reduce into one of the already used methods in Table 1 or becomes a new metric that needs further justification.

**Reasons To Accept:**

This paper addresses an important problem, especially now when there are more ties than ever due to higher quality generation models such as LLMs. The paper is well written, easy to follow (thanks to the toy examples), with sound evidence. The proposed meta-evaluation method is easy to incorporate in shared tasks (such as WMT metrics), thus this could lead to a real change in MT meta evaluation community.


**Reasons To Reject:**

We donot see any manor reasons. If we want to try very hard -- the empirical results are based on 3 language pairs (en-de, zh-en, en-ru on WMT22). Perhaps it'd be interested to see how this works with other languages and annotation methods that are not MQM.


**Reproducibility:**

5: Could easily reproduce the results.

**Reviewer Confidence:**

4: Quite sure. I tried to check the important points carefully. It's unlikely, though conceivable, that I missed something that should affect my ratings.

---

> ### Author Rebuttal · Authors · 2023-08-24
>
> Thanks so much for your review of our paper.
>
> Regarding your question about when there are no ties: Thanks for this great question. Assuming you mean there are no ties in the human scores, then our proposal is equivalent to tau_10. Importantly, however, when tau_10 was used by WMT, they intentionally removed pairs that were tied according to the human, which we argue is an incorrect thing to do.
>
> We believe that the tau_eq and acc_eq formulas are already justified in the absence of ties in the human and/or metric scores. They still compute an agreement between the metric and human scores when rewarding correct tie predictions. If there are no human ties, tie calibration will not introduce any ties because any ties will be incorrect and existing ties will correctly be penalized because the human did not believe the pair to be tied, which we believe is justified.

---

### Meta-Review · Area_Chair_nTp2 · 2023-09-02

**Recommendation:** 5
**Confidence:** 4
**Best Paper Recommendation:** No

**Metareview:**

The paper shows the failures of Kendall’s tau in presence of ties when evaluation MT evaluation metrics scores and proposes a remedy. The proposed metric introduces predictions of ties and rewards them. With the increase of quality, ties become more frequent. Also ties in WMT have become more reliable, and frequent with the MQM HE scores (#of errors). Some variants of Tau discard ties loosing important information. The analysis of the calibration mechanism for regression-based metrics is very informative, the calibration method does not generalize across dissimilar data sets.
Reasons to accept: paper  discovers and proposes a fix to relevant problem in MT meta-evaluation. The paper is well written and could have a major impact in the community leading to fairer comparisons across metrics.
Reasons to reject: Limited language pairs are analyzed but they are very diverse. Evaluation limited to MQM HE framework, which is understandable.

**Meta-Review:**

Very clearly written paper which addresses and fixes a failure of a very used metric for evaluating MT quality scores.
The approach is sound and the analysis is convincing.

---

### Decision · Program_Chairs · 2023-10-07

**Decision:**

Accept-Main

**Comment:**

The paper shows the failures of Kendall’s tau in presence of ties when evaluation MT evaluation metrics scores and proposes a remedy. The proposed metric introduces predictions of ties and rewards them. With the increase of quality, ties become more frequent. Also ties in WMT have become more reliable, and frequent with the MQM HE scores (#of errors). Some variants of Tau discard ties loosing important information. The analysis of the calibration mechanism for regression-based metrics is very informative, the calibration method does not generalize across dissimilar data sets.
Reasons to accept: paper  discovers and proposes a fix to relevant problem in MT meta-evaluation. The paper is well written and could have a major impact in the community leading to fairer comparisons across metrics.
Reasons to reject: Limited language pairs are analyzed but they are very diverse. Evaluation limited to MQM HE framework, which is understandable.